# DePass: Unified Feature Attributing by Simple Decomposed Forward Pass

**Xiangyu Hong**[1]*, **Che Jiang**[1]*, **Kai Tian**[1], **Biqing Qi**[2], **Youbang Sun**[1], **Ning Ding**[1,2], **Bowen Zhou**[1,2]†

[1] Department of Electronic Engineering, Tsinghua University
[2] Shanghai AI Laboratory
`hong-xy22@mails.tsinghua.edu.cn jc23@mails.tsinghua.edu.cn`
`zhoubowen@tsinghua.edu.cn`

## Abstract

Attributing the behavior of Transformer models to internal computations is a central challenge in mechanistic interpretability. We introduce DePass, a unified framework for feature attribution based on a single decomposed forward pass. DePass decomposes hidden states into customized additive components, then propagates them with attention scores and MLP's activations fixed. It achieves faithful, fine-grained attribution without requiring auxiliary training. We validate DePass across token-level, model component-level, and subspace-level attribution tasks, demonstrating its effectiveness and fidelity. Our experiments highlight its potential to attribute information flow between arbitrary components of a Transformer model. We hope DePass serves as a foundational tool for broader applications in interpretability. Code is available at `https://github.com/TsinghuaC3I/Decomposed-Forward-Pass`

## 1 Introduction

Mechanistic interpretability is the foundation to monitor, modify, and predict Transformer-based models' behavior. The first step of such reverse engineering is to decompose the neural network and analyze what contributes to the model's behavior [1, 2]. Researchers continue to develop methods for decomposing this highly complex system. Without modifying or abstracting the network, directly applying noise ablations [3, 4] or activation patching [5–7] to all modules is computationally expensive and provides limited insight into intermediate information flow [8]. Gradient-based attribution methods also face theoretical challenges [9]. In contrast, approximating or abstracting the model can partially align with human cognition [10–13], but often fails to reach fine-grained components such as neurons or attention heads. Moreover, non-conservative approximations may compromise the faithfulness of attribution. In this paper, we propose DePass (**De**composed Forward **Pass**), a direct and unified feature attribution framework that addresses these shortcomings through a single decomposed forward pass.

The main concept of DePass is simple: we break down every hidden state into additive components, propagate these components through the remaining layers, and then obtain each component's exact contribution to the target representation. In the decomposed forward pass, attention scores and MLP activations are fixed, and weighted contributions are assigned based on the decomposed components. This method guarantees several advantages:

- **Faithfulness and Completeness**: By freezing attention scores and MLP activations, DePass eliminates second-order effects when propagating forward [8]. As a result, per-component summation reconstructs exactly the hidden state of the original model.

---

*Equal contribution.
†Corresponding author.

39th Conference on Neural Information Processing Systems (NeurIPS 2025).

- **Unification Across Components**: DePass provides a unified attribution framework for multiple model granularities—including input tokens, attention heads, neurons, and subspaces of the residual stream. This enables consistent, fine-grained interpretability without modifying the model or using task-specific approximations.

- **Representation-Level Attribution**: Unlike conventional attribution scores [11], DePass tracks how decomposed representations evolve through the forward pass. This facilitates more fine-grained attributions and a natural interface to align with human reasoning and sparse dictionary learning (SDL) such as SAE.

We conduct attribution experiments at the token level, model component level, and subspace level to evaluate the effectiveness and generality of DePass. Our results show that DePass enables lossless, additive decomposition of hidden states throughout the forward pass of Transformer models, allowing faithful tracking of information flow according to attribution needs. This unified framework offers a new analytical tool for mechanistic interpretability.

## 2 Architecture of Transformer-Based Autoregressive Language Models

Transformer-based autoregressive language models use the residual stream as the backbone for information processing. In DePass, the decomposed components follow the same pathway as the original hidden states, passing through each decoder layer's multi-head self-attention (MHSA) sublayer, feedforward network (MLP) sublayer, and layer normalization (LayerNorm) operations. In this section, we establish the notations used in the standard forward pass, laying the groundwork for the formal definition of the DePass method.

### 2.1 Transformer Decoder Layers

Let $X^\ell \in \mathbb{R}^{N \times D}$ denote the token representations at layer $\ell$, where $N$ is the sequence length and $D$ is the dimension of model's hidden state.

**Multi-Head Self-Attention (MHSA)**   The multi-head self-attention (MHSA) mechanism enables each token to attend to others in the sequence, capturing contextual dependencies. At layer $\ell$, the input $X^{\ell-1} \in \mathbb{R}^{N \times D}$ is first normalized by layer normalization, $\tilde{X}^\ell = \mathrm{LN}(X^{\ell-1})$.

For each head $j = 1, \ldots, H$, queries, keys, and values are computed by linear projections of $\tilde{X}^\ell$, and attention scores are obtained via scaled dot-product attention with a causal mask $M^{\ell,j}$:

$$A^{\ell,j} = \mathrm{Softmax}\left( \frac{Q^{\ell,j}(K^{\ell,j})^\top}{\sqrt{D/H}} + M^{\ell,j} \right). \tag{1}$$

Letting $W_{VO}^{\ell,j} = W_V^{\ell,j} W_O^{\ell,j} \in \mathbb{R}^{D \times D}$, the output of all attention heads is:

$$Output_{\mathrm{attn}}^\ell = \sum_{j=1}^{H} O^{\ell,j}, \quad \text{where } O^{\ell,j} = A^{\ell,j} \tilde{X}^\ell W_{VO}^{\ell,j}. \tag{2}$$

Finally, the residual connection is applied:

$$X_{\mathrm{attn}}^\ell = X^{\ell-1} + Output_{\mathrm{attn}}^\ell. \tag{3}$$

**Feedforward Network (MLP)**   After the attention sublayer, hidden states are passed through a position-wise feedforward network to enhance per-token representation capacity. In particular, a LayerNorm is applied to the attention output, and then the MLP operates independently at each position $i$:

$$Output_{\mathrm{ffn}}^\ell = W_D^\ell\, \sigma\left( W_U^\ell\, \mathrm{LN}(X_{\mathrm{attn}}^\ell) \right), \tag{4}$$

where $W_U^\ell \in \mathbb{R}^{d_{\mathrm{ffn}} \times D}$, $W_D^\ell \in \mathbb{R}^{D \times d_{\mathrm{ffn}}}$, and $\sigma(\cdot)$ is a nonlinear activation function. MLPs can be regarded as a soft key-value retrieval mechanism [14]. Let

$$W_U^\ell = \begin{bmatrix} \mathbf{f}_1^\ell \\ \vdots \\ \mathbf{f}_N^\ell \end{bmatrix} \in \mathbb{R}^{N \times D}, \quad W_D^\ell = \begin{bmatrix} \mathbf{v}_1^\ell & \cdots & \mathbf{v}_N^\ell \end{bmatrix} \in \mathbb{R}^{D \times N}, \tag{5}$$

where $\mathbf{f}_k^\ell \in \mathbb{R}^D$ and $\mathbf{v}_k^\ell \in \mathbb{R}^D$ represent the $k$-th subkey and subvalue. The output at position $i$ is:

$$Output_{\text{ffn},i}^\ell = \sum_{k=1}^N m_{i,k}^\ell \, \mathbf{v}_k^\ell, \quad m_{i,k}^\ell = \sigma\left(\mathbf{f}_k^\ell \cdot \tilde{X}_{\text{attn},i}^\ell\right), \tag{6}$$

where $\tilde{X}_{\text{attn},i}^\ell = \text{LN}(X_{\text{attn},i}^\ell)$ acts as a query vector over the subkeys.

Finally, the residual connection yields the output for the next layer:

$$X^{\ell+1} = X_{\text{attn}}^\ell + Output_{\text{ffn}}^\ell. \tag{7}$$

## 2.2 Language Modeling Head

At position $i$, the final hidden state $X_i^L \in \mathbb{R}^D$ is normalized and projected by language modeling head $W_{\text{lm}} \in \mathbb{R}^{V \times D}$, where $V$ is the vocabulary size. The resulting logits yield the next-token distribution:

$$P(w_{i+1} \mid w_{\leq i}) = \text{Softmax}\left(W_{\text{lm}} \, \text{LN}(X_i^L)\right), \tag{8}$$

where $w_{i+1}$ is the predicted token at the next position..

# 3 Decomposed Forward Pass

We introduce DePass (Decomposed Forward Pass), a method for isolating and tracing representational components through Transformer layers. This section outlines: (1) the initialization of decomposed hidden states; (2) the reformulated forward pass for propagating these components; and (3) the use of decomposed states for decoding and attribution. Throughout this paper, we use the notation $X[i_1, i_2, \ldots]$ to index into multidimensional tensors following the NumPy-style indexing convention.

## 3.1 Initialization of Decomposed Hidden States

Given a hidden state $X^{(\ell)} \in \mathbb{R}^{N \times D}$ at layer $\ell$, with $N$ tokens and hidden dimension $D$, we define its decomposed form $X_{\text{dec}}^{(\ell)} \in \mathbb{R}^{N \times M \times D}$, where $M$ is the number of decomposition components. Each token representation is the sum of its components:

$$X_i^{(\ell)} = \sum_{m=1}^M X_{\text{dec}}^{(\ell)}[i, m, :]. \tag{9}$$

The index $m$ is shared across positions to ensure semantic alignment, and its interpretation depends on task-specific decomposition strategies.

## 3.2 Propagating Decomposed Hidden States Through Transformer Blocks

We describe how decomposed hidden states propagate through LayerNorm, Multi-Head Self-Attention (MHSA), and MLP modules in a Transformer block.

**LayerNorm** Since most layer normalization performs feature-wise scaling and shifting, we illustrate here how the more efficient RMSNorm—used in our experimental models—operates on decomposed hidden states. Given token $i$'s hidden state $X_i^{(\ell)} = \sum_{m=1}^M X_{\text{dec}}^{(\ell)}[i, m, :]$, RMSNorm computes:

$$\text{RMSNorm}(X_i^{(\ell)}) = \gamma \odot \frac{X_i^{(\ell)}}{\text{RMS}(X_i^{(\ell)})}, \quad \text{where } \text{RMS}(X_i^{(\ell)}) = \sqrt{\frac{1}{d}\left\|X_i^{(\ell)}\right\|^2}. \tag{10}$$

The scaling operation in RMSNorm is additive over decomposed components, allowing the scaling factor to be applied independently to each component, so it distributes over components as:

$$\tilde{X}_{\text{dec}}^{(\ell)}[i, m, :] = A_i X_{\text{dec}}^{(\ell)}[i, m, :], \quad \text{where } A_i = \text{diag}\left(\frac{\gamma}{\text{RMS}(X_i^{(\ell)})}\right). \tag{11}$$

**Multi-Head Self-Attention (MHSA) on Decomposed Hidden States**  Let $X_{\text{dec}}^{(\ell-1)} \in \mathbb{R}^{N \times M \times D}$ represent the decomposed hidden states from the previous layer. To apply multi-head self-attention in a component-aware manner, we extend the standard attention formulation (see Eq. 2) to operate on each component individually. For each attention head $j$, the attention output for token $i$ from component $m$ is:

$$\mathbf{o}_{i,m}^{(\ell,j)} = A^{(\ell,j)} X_{\text{dec}}^{(\ell-1)}[i,m,:] W_{VO}^{(\ell,j)}, \tag{12}$$

where $A^{(\ell,j)}$ is the original attention scores. The MHSA output is aggregated across all heads and components, combined with residuals:

$$X_{\text{attn,dec}}^{(\ell)}[i,m,:] = X_{\text{dec}}^{(\ell-1)}[i,m,:] + \sum_{j=1}^{H} \mathbf{o}_{i,m}^{(\ell,j)}. \tag{13}$$

**MLP on Decomposed Hidden States**  The MLP module processes decomposed representations $X_{\text{attn,dec}}^{(\ell)} \in \mathbb{R}^{N \times M \times D}$ by distributing each neuron's output across components. For each token $i$, component $m$, and neuron $k$, we compute a relevance score:

$$a_{i,m,k} = \left(\mathbf{f}_k^\ell\right)^\top X_{\text{attn,dec}}^{(\ell)}[i,m,:], \quad \alpha_{i,m,k} = \frac{\exp(a_{i,m,k})}{\sum_{m'=1}^{M} \exp(a_{i,m',k})}, \tag{14}$$

where $\alpha_{i,m,k}$ determines how neuron $k$'s output is apportioned to component $m$. Alternative normalization methods to softmax are compared in Appendix A.

The updated decomposed hidden state is then:

$$X_{\text{dec}}^{(\ell+1)}[i,m,:] = X_{\text{attn,dec}}^{(\ell)}[i,m,:] + \sum_{k=1}^{D_{\text{mlp}}} \alpha_{i,m,k} \cdot m_{i,k}^\ell \cdot \mathbf{v}_k^\ell, \tag{15}$$

where $m_{i,k}^\ell$ is the neuron activation and $\mathbf{v}_k^\ell \in \mathbb{R}^D$ is the output projection. The original hidden state is recovered by summing over components:

$$X_i^{(\ell+1)} = \sum_{m=1}^{M} X_{\text{dec}}^{(\ell+1)}[i,m,:]. \tag{16}$$

Given the decomposed hidden states from the previous layer, each component can be further propagated through the next layer independently. This maintains disentangled contributions across components and allows exact reconstruction of the original hidden state. All computations are fully parallelizable, incurring minimal overhead.

### 3.3  LM Head and Attribution Score

Let $X_i^{(L)} = \sum_{m=1}^{M} X_{\text{dec}}^{(L)}[i,m,:]$ be the final hidden state of token $i$, and let $\mathbf{w}_y = W_{\text{LM}}[y,:]$ be the LM head vector for target $y$. The output logit and component-wise attribution are:

$$\text{logits}_y = \mathbf{w}_y^\top X_i^{(L)}, \quad \Delta\text{logits}_{y,m} = \mathbf{w}_y^\top X_{\text{dec}}^{(L)}[i,m,:], \tag{17}$$

with $\sum_{m=1}^{M} \Delta\text{logits}_{y,m} = \text{logits}_y$. This yields fine-grained attribution across components. To attribute with respect to a given subspace at a particular layer, we project each $X_{\text{dec}}^{(\ell)}[i,m,:]$ onto its direction and use the projection values as scores.

## 4  DePass Attribution across multi-granular Levels

### 4.1  Token-Wise DePass

**Token-Wise Decomposed Hidden States Initialization**  To analyze how each input token contributes to the model's hidden states, we initialize a token-wise decomposition of the hidden states: each token's hidden state is split into $N$ additive components, one per input token. At the embedding layer, the decomposition is defined as:

$$X_{\text{dec}}^{(0)}[i,m,:] = \begin{cases} X^{(0)}[i,:], & \text{if } i = m \\ 0, & \text{otherwise} \end{cases} \tag{18}$$

where $X^{(0)} \in \mathbb{R}^{N \times d}$ is the input embedding and $X_{\text{dec}}^{(0)} \in \mathbb{R}^{N \times N \times d}$ is its token-level decomposition.

This structure is propagated layer by layer according to the method in Section 3.2. At any layer, the $m$-th component of the decomposed hidden state represents the contribution of the $m$-th input token to the overall hidden states.

### 4.1.1 Token-Level Output Attribution via DePass

**Problem Definition.** Given a model $\mathcal{M}$, input $x = [x_1, \ldots, x_n]$, and output $\hat{y} = \mathcal{M}(x)$, token-wise attribution assigns each token $x_i$ a score $s_i \in \mathbb{R}$ indicating its influence on $\hat{y}$—with higher scores denoting greater impact.

**DePass-Based Output Attribution.** Starting from token-wise decomposition initialization (Eq. 18) and applying the forward pass decomposition (Section 3.2), we use the language modeling head (Section 3.3) on the final decomposed hidden states. This yields token-level attribution scores, quantifying how each input token contributes to the model's prediction of output $y$.

**Experiment Setup. Baselines.** We compare against standard attribution methods on a fixed pretrained model. **Gradient-based:** Input×Gradient [15], Integrated Gradients [16], Gradient SHAP [17]; **Attention-based:** Mean Attention, Last-layer Attention [18], and Attention Rollout [6].

**Tasks.** We evaluate on two benchmarks targeting different reasoning types: **Known_1000** [4][3] (factual QA, e.g., "Audible.com is owned by") and **IOI** [19] (indirect object identification, e.g., "Eleanor and Deanna were thinking about going to the mountain. Eleanor wanted to give a watermelon to").

**Evaluation Protocol.** For each input $x$, we first compute attribution scores for the correct answer via various methods. Based on these scores, we apply token-level interventions: **patch top**—mask the top $K\%$ tokens with highest attribution; **recover top**—mask the bottom $(100-K)\%$ tokens (lowest attribution) and then restore the top $K\%$.

The remaining tokens are reassembled into a new prompt and fed to the model. Faithfulness is then evaluated by measuring the change in the predicted probability of the correct answer: **Comprehensiveness:** Drop in probability under patch top (higher is better); **Sufficiency:** Retained probability under recover top (lower is better).

We compute the relative change in predicted probability as:

$$\Delta p^{(K)} = \left| \frac{p(\hat{y} \mid x) - p(\hat{y} \mid \tilde{x}^{(K)})}{p(\hat{y} \mid x)} \right|,$$

where $\hat{y}$ denotes the target token and $\tilde{x}^{(K)}$ denotes the perturbed input under each intervention. The figures report the average $\Delta p^{(K)}$ across all data points per dataset at each masking level.

To ensure a fair comparison with other token-level baselines, in the experiments corresponding to Figure 1, we perform ranking and ablation at the subword token level. DePass also supports word-level attribution, with examples illustrated in Appendix B.3.

**Results.** As shown in Figure 1, our method yields substantially higher **comprehensiveness** scores on both Known_1000 and IOI benchmarks using Llama-2-13b-chat-hf, indicating that the tokens identified by DePass are more critical to the model's prediction.

In terms of **sufficiency**, all methods are comparable. This is partly due to evaluation limits: when keeping only top tokens, most methods recover those key for prediction. Attention-based methods, which preserve broad semantic structure, perform slightly better. A representative case of output attribution is shown in the top of Figure 2.

Additional results on other model variants and experiment details are provided in Appendix B.

### 4.1.2 Token-level Subspace Attribution via DePass

While methods such as probing or sparse autoencoders (SAEs) are able to uncover meaningful subspaces, they fall short of directly identifying which input tokens are responsible for activating

---

[3]Dataset can be found at: `https://rome.baulab.info/data/dsets/known_1000.json`

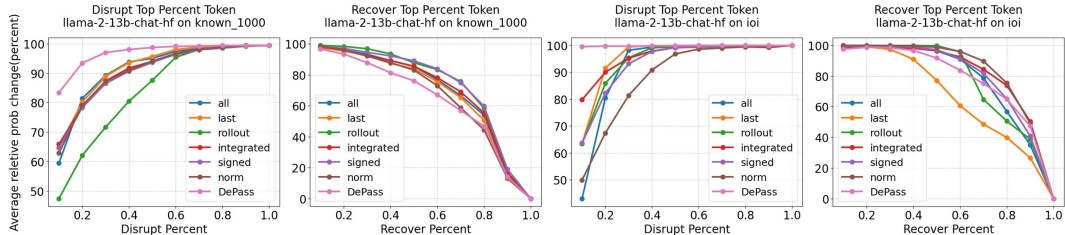

Figure 1: Faithfulness evaluation on Known_1000 and IOI using Llama-2-13b-chat-hf. Our method yields better comprehensiveness and competitive sufficiency.

those subspaces. Existing approaches that attempt to link tokens with feature activations [20] often depend on intricate graph construction and pruning, making them computationally expensive and difficult to scale. To address this limitation, we introduce DePass as a framework that attributes subspace activations to specific input tokens in a more direct and efficient manner.

**DePass-Based Subspace Attribution.** Given a semantic direction $\mathbf{v} \in \mathbb{R}^D$ and decomposed hidden states $X_{\text{dec}}^{(\ell)} \in \mathbb{R}^{N \times N \times D}$ at layer $\ell$, the subspace activation at position $i$ is:

$$a_i = \mathbf{v}^\top X_i^{(\ell)} = \sum_{m=1}^{N} \mathbf{v}^\top X_{\text{dec}}^{(\ell)}[i, m, :], \tag{19}$$

where each term quantifies the contribution of token $x_m$ to the subspace activation at position $i$. This enables fine-grained attribution to arbitrary linear subspaces.

**Experiment: Probing Subspace Attribution via DePass** We evaluate our method in the context of factuality, where prior work has shown that truthfulness can be linearly separated in hidden states [21]. A linear probe $f(x) = \mathbf{w}^\top x + b$ is trained to detect untruthful activations, with $\mathbf{w}$ defining a factuality subspace. Some methods leverage such probes to mask misleading input tokens based on whether their intermediate hidden states align with untruthful directions [22]. In contrast, our method attributes subspace activations directly to specific input tokens, enabling more fine-grained interventions such as selectively masking misleading content.

**Evaluation Setup.** We evaluate on two factuality benchmarks: **CounterFact[4]** (modified for generation, prompting completions to factual statements) and **TruthfulQA[23]** (converted to multiple-choice with misleading options).

We compare the effectiveness of DePass and TACS[22] in identifying misleading tokens that contribute to model errors. Each example is tested under four settings: (1) **No Information**: given only the question, (2) **Misinformation**: given wrong information along with the question, (3) **Misinformation + TACS Masking [22]**: tokens aligned with untruthful directions are masked, and (4) **Misinformation + Ours (DePass Masking)**: tokens contributing most to untruthful activation are masked. For each prompt, both methods mask the same number of tokens based on classifier confidence.

**Results.** As shown in Table 1, DePass-based masking consistently improves factual accuracy across models and datasets. Compared to direct probing-based masking, our method yields stronger gains. For example, on Llama-2-7b-chat-hf, accuracy rises from 10.16% (misinformation) to 43.13% (ours), demonstrating DePass's ability to isolate and suppress harmful inputs. A representative case of subspace attribution is shown in the bottom of Figure 2.

See Appendix C for details on dataset formatting, probe training, prompt examples, and our strategy for selecting masked tokens based on classifier outputs from different layers. We also provide examples of applying DePass for SAE feature attribution to illustrate the potential of combining the two methods (Appendix C.5).

Table 1: Accuracy (%) across different input settings on CounterFact and TruthfulQA. Masking is applied to 30% of misleading tokens identified using either prior methods or our DePass-based attribution.

| Model | Dataset | No Info | Misinformation | + TACS Masking | + Ours (DePass Masking) |
|---|---|---|---|---|---|
| Llama-2-7b-chat-hf | CounterFact (Gen) | 57.52 | 10.16 | 25.68 | **43.13** |
| | TruthfulQA (MC) | 66.10 | 33.05 | 43.57 | **46.51** |
| Llama-2-13b-chat-hf | CounterFact (Gen) | 60.68 | 4.88 | 13.06 | **34.90** |
| | TruthfulQA (MC) | 73.56 | 38.92 | 49.33 | **53.00** |
| Llama-3.1-8b-Instruct | CounterFact (Gen) | 60.63 | 3.30 | 16.03 | **59.16** |
| | TruthfulQA (MC) | 83.60 | 70.26 | 71.48 | **76.62** |
| Qwen2-1.5B-Instruct | CounterFact (Gen) | 44.04 | 3.51 | 17.75 | **43.32** |
| | TruthfulQA (MC) | 71.24 | 46.14 | **60.10** | 51.90 |
| Qwen2-7B-Instruct | CounterFact (Gen) | 39.43 | 6.54 | 23.91 | **29.68** |
| | TruthfulQA (MC) | 77.23 | 47.25 | **68.42** | 64.87 |
| Meta-Llama-3.1-70B-Instruct | CounterFact (Gen) | 72.20 | 9.51 | 33.29 | **55.62** |
| | TruthfulQA (MC) | 89.96 | 65.61 | 74.54 | **76.87** |

Attribution Scores for Model Output: "Germany"

|  | The | artist | lives | in | Berlin | , | the | capital | of |
|---|---|---|---|---|---|---|---|---|---|
| 0.01 | 1.03 | 1.14 | 0.70 | 0.56 | 8.48 | 0.55 | 0.51 | 5.27 | 0.48 |

Average Attribution Scores Across Layers for the Truthful Classifier

|  | _The | _artist | _lives | _in | _Berlin | , | _the | _capital | _of | _France |
|---|---|---|---|---|---|---|---|---|---|---|
| 1.29 | -2.00 | -0.79 | -0.89 | -0.64 | -2.02 | -0.29 | -0.25 | -8.25 | -0.38 | -3.13 |

Figure 2: Two input-level attribution examples. The top case illustrates contributions to the model prediction "Germany", with higher scores indicating greater influence. The bottom case shows token-wise contributions to the classifier's prediction of the "truthful" label, where more negative scores support the untruthful classification (label 0).

## 4.2 Model Component-Wise DePass

**Model Component-Wise Decomposed Hidden States Initialization**  To attribute model behavior to specific architectural components, we perform component-level decomposition of hidden states, targeting attention heads and MLP neurons.

**Attention Head:** For a Transformer layer with $H$ attention heads, we assign one decomposition component to each head and one to the residual connection, resulting in $M = H + 1$. For token $i$, we define:

$$X_{\text{dec}}^{(\ell)}[i, h, :] = \mathbf{o}_i^{(\ell, h)}, \quad h = 1, \ldots, H; \quad X_{\text{dec}}^{(\ell)}[i, H+1, :] = X^{(\ell-1)}[i, :] \tag{20}$$

where $\mathbf{o}_i^{(\ell, h)}$ is the output of the $h$-th attention head at layer $\ell$.

**MLP Neurons:** Similarly, for an MLP block with $N_{\text{MLP}}$ hidden neurons, we decompose the output and residual connection into $N_{\text{MLP}} + 1$ components:

$$X_{\text{dec}}^{(\ell)}[i, n, :] = m_{i,n}^{\ell} \mathbf{v}_n^{\ell}, \quad X_{\text{dec}}^{(\ell)}[i, N_{\text{MLP}}+1, :] = X_{\text{attn}}^{\ell}[i, :]$$

where $m_{i,n}^{\ell} \mathbf{v}_n^{\ell} \in \mathbb{R}^D$ is the contribution of the $n$-th MLP neuron to the hidden states of token $i$.

This setup aligns with our decomposition-based forward pass and enables localized interpretability. Component-wise importance scores for a given output are computed as described in §3.3.

**Experiment: Evaluating Component Importance via Masking.**  We validate the functional significance of components using masking-based ablations guided by various importance metrics.

**Importance Scoring Methods.**  We compare several scoring strategies: **Norm:** $\ell_2$ norm of each component's activation; **Coef:** Absolute activations after MLP up-projection and nonlinearity; used only for MLP neurons. **AtP** [24] (*Activation × Gradient*): A gradient-based importance measure using first-order Taylor approximation; **DePass (Ours):** Attribution scores derived from decomposed hidden states; **DePass-Abs:** Absolute values of DePass scores, capturing both supportive and suppressive contributions.

**Tasks and Setup.** Experiments are conducted on: **IOI (Indirect Object Identification):** Synthetic reasoning benchmark; **CounterFact (QA):** Factual knowledge recall task. Only correctly answered examples are used for meaningful attribution.

**Evaluation Protocol.** We perform two complementary masking interventions: **Top-$k$ Masking (Comprehensiveness):** Mask the top-$k$ components, with a sharp accuracy drop indicating critical components. **Bottom-$k$ Masking (Sufficiency):** Mask only the bottom-$k$ components; high accuracy retention suggests sufficiency for prediction.

Masking is applied structurally: attention heads are ablated by zeroing their output projections $W_{VO}^{(\ell,h)}$, and MLP neurons by zeroing their activations before projection. We report average accuracy across examples and for a robust attribution quality assessment.

**Results** Figure 3 presents the results on Llama-2-7b-chat-hf; additional results for other models are provided in the Appendix D. Our method consistently outperforms baseline attribution techniques across different masking strategies. In particular, it achieves a more pronounced accuracy drop in Top-$k$ Masking and better accuracy retention in Bottom-$k$ Masking, indicating its superior ability to identify and attribute critical components.

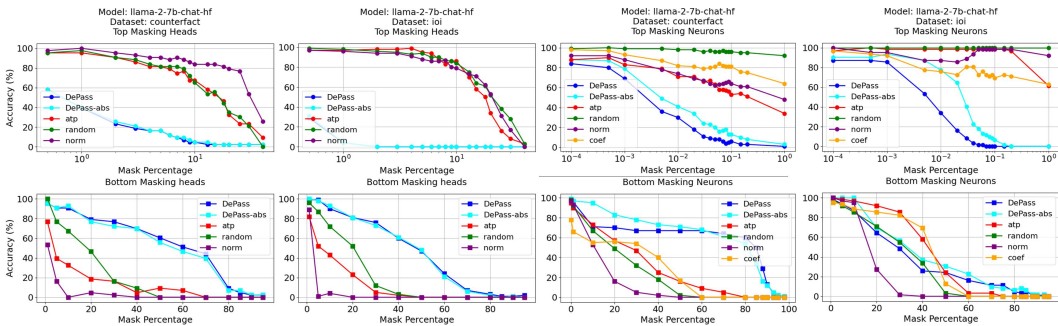

Figure 3: Performance of our method compared to baseline attribution techniques on Llama-2-7b-chat-hf under various masking strategies. Our approach more accurately identifies critical components, as reflected by the sharper drop in Top-$k$ Masking and stronger performance in Bottom-$k$ Masking.

### 4.3 Subspace-Level Decomposition

**Subspace-Wise Decomposition of Hidden States** To analyze how specific representational directions affect model behavior, we decompose hidden states at layer $\ell$ into components within and orthogonal to a target subspace. Given a projection matrix $\mathbf{P}_t \in \mathbb{R}^{D \times D}$, each token's hidden state $X^{(\ell)}[i,:] \in \mathbb{R}^d$ is split as:

$$X_{\text{dec}}^{(\ell)}[i,0,:] = \mathbf{P}_t X^{(\ell)}[i,:], \quad X_{\text{dec}}^{(\ell)}[i,1,:] = (\mathbf{I} - \mathbf{P}_t)X^{(\ell)}[i,:], \tag{21}$$

where $\mathbf{I}$ is the identity matrix. These components are then independently propagated using DePass (Section 3.2), allowing attribution of model behavior to the subspace. Construction details of $\mathbf{P}_t$ are provided in Appendix E.2.

**Experiment: Interpreting Language Subspace Effects** We examine whether the decomposition framework can reveal functionally distinct subspaces in multilingual settings. Specifically, we hypothesize that model hidden states can be separated into a *language subspace* capturing language signals, and a *semantic subspace* encoding language-invariant meaning[25, 26].

**Subspace Construction.** We apply DePass to Llama-3.1-8B-Instruct, a multilingual model trained on English, French, German, and more. We train a token-level language classifier on *CounterFact* [4], translated into multiple languages, to detect the language of each input token. The classifier weights define a *language-diagnostic subspace*, and its orthogonal complement serves as the *semantic subspace*. Token hidden states are projected into these two subspaces as described in Section 4.3, and decomposition is propagated forward through the model.

**Evaluation Setup.** To evaluate whether DePass faithfully attributes model behavior to distinct subspaces, we begin by projecting hidden states at an intermediate layer (layer 15) into the *language* and *semantic* subspaces. These decomposed components are then independently propagated through the remaining layers of the model, yielding two final representations at the last layer: $X_{\text{lang}}^{\text{dec}}[i] = X_{\text{dec}}^{(L)}[i, 0, :]$ and $X_{\text{sem}}^{\text{dec}}[i] = X_{\text{dec}}^{(L)}[i, 1, :]$.

We decode these representations separately by applying the language modeling head to each of them. Table 2 reports the top-5 tokens generated from $X_{\text{lang}}^{\text{dec}}$ and $X_{\text{sem}}^{\text{dec}}$ for multilingual prompts (e.g., "What is Thomas Joannes Stieltjes's native language? It is").

**Results.** Tokens generated from the **semantic subspace** ($X_{\text{sem}}^{\text{dec}}$) consistently reflect factual content (e.g., "Dutch", "Holland"), largely invariant to the input language. In contrast, the **language subspace** ($X_{\text{lang}}^{\text{dec}}$) produces lexical or stylistic tokens (e.g., "né", "nicht", "de") aligned with the prompt's language. These results show that DePass faithfully propagates and preserves the functional roles of each subspace, enabling clear attribution of linguistic and semantic behavior and highlighting its potential for subspace-level analysis.

To further verify that DePass faithfully attributes language-related behavior, we apply t-SNE to $X_{\text{lang}}^{\text{dec}}$ from multilingual inputs. As shown in Figure 4, the resulting clusters align with language identity, confirming that DePass effectively preserves and propagates language-specific signals. These results highlight its effectiveness for subspace-level attribution. Additional examples are in Appendix E.

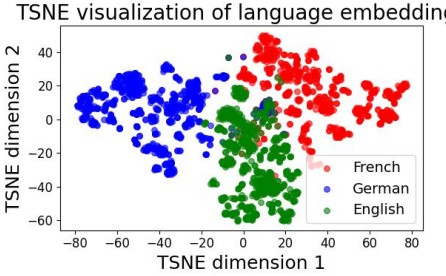

Figure 4: t-SNE visualization of token-wise projections onto the language subspace. Distinct clusters indicate strong language-specific structure.

| Language | Language Tokens |
|---|---|
| English | a, the, an, not, N |
| French | né, consid, de, conn, ét |
| German | nicht, keine, eine, die, das |

| Language | Semantic Tokens |
|---|---|
| English | Dutch, dut, reported, said, Afrika |
| French | Dutch, holland, Holland, of, N |
| German | Dutch, Holl, Hol, Holland, also |

Table 2: Top-5 tokens decoded from the `language` and `semantic` subspaces for different multilingual prompts.

## 5 Related Work

**Decomposition-based Attribution.** Without modifying or approximating Transformer modules, decomposing hidden states can partially explain model behavior [27]. Decomposition of MLPs and attention blocks at the logit level enables attribution of their residual contributions to model's output [28–31]. Additionally, attention scores have been used to construct information flow graphs that trace input tokens' influence [6, 11, 32]. Combining decomposition with gradient information can further highlight salient features [15, 17, 16, 33]. DecompX [34] applies similar decomposition ideas in restricted settings; our approach generalizes this to arbitrary modules in generative Transformers with finer-grained decomposition of MLP activations. Compared to these methods, DePass performs a more direct additive decomposition of the computation, avoiding early mapping to scalar saliency scores and thereby preserving both faithfulness and interpretability.

**Ablation-based Attribution.** Another line of work attributes importance by ablating specific components—such as adding noise or zeroing out input tokens or internal activations—and measuring the resulting change in the output distribution [4, 3, 35–37]. Since exhaustively ablating all components incurs high computational cost, approximate methods or surrogate models have been proposed to accelerate activation patching [38, 12, 39]. In contrast, DePass provides more faithful attribution through direct decomposition, avoiding the indirectness and potential artifacts of ablation-based approaches.

**Sparse-Dictionary-Learning-based Attribution.** Sparse Dictionary Learning (SDL) is currently the most popular decomposition methods in interpretability. Based on the superposition hypothesis [40, 41], SDL aims to recover more interpretable components than the original feature dimensionality. SAE-based methods supervise the reconstruction of current activations [42–45], while Transcoder [20] targets the reconstruction of next-layer activations, and CrossCoder[46] extends this to jointly reconstruct activations across multiple layers. However, the high training cost [1], annotation effort, and reconstruction errors [47] limit their scalability. Some approaches aim to track feature evolution during forward propagation [48–50], but they often rely on proxies like inter-feature correlation or similarity scores rather than precisely tracing the actual transformations of features. Nevertheless, SAE remains valuable for improving downstream task performance (such as model steering [51]) and can be used effectively in conjunction with DePass to bridge model representations and human-interpretable concepts.

## 6    Conclusions

In this paper, we present DePass, a simple yet efficient framework for interpreting Transformer models via decomposed forward pass. By freezing and allocate attention scores and MLP activations, DePass enables lossless additive decomposition, and can be applied to any Transformer-based architecture. DePass achieves more faithful attribution across different levels of granularity comparing to other methods. We hope DePass serves as a general-purpose tool for mechanistic interpretability and inspires broader adoption and diverse applications across the community.

## 7    Acknowledgements

This work is supported by the National Science and Technology Major Project (2023ZD0121403), Young Elite Scientists Sponsorship Program by CAST (2023QNRC001), National Natural Science Foundation of China (No.62406165), Shanghai Municipal Science and Technology Major Project, and the Beijing Natural Science Foundation Undergraduate "Qiyan Research Program" (No. QY24259). We extend our gratitude to the anonymous reviewers for their insightful feedback, which has greatly contributed to the improvement of this paper.

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

# A Empirical Comparison of Softmax and Alternative Functions for MLP Attribution

To assess the design choice of using softmax for MLP attribution (Eq. 14), we compare it against two alternative normalization methods: (1) **Linear-norm normalization**: subtracting the minimum and dividing by the sum; (2) **Linear-weighted decomposition**: linearly decomposing the contribution to the original activation value for each element. This comparison assesses whether the softmax rule is empirically justified.

We report results on the `Known_1000` dataset using the `llama-2-7b-chat-hf` model for both **patch-top** and **recover-top** token removal strategies.

| Patch-Top (%) | 0.1 | 0.2 | 0.3 | 0.4 | 0.5 | 0.6 | 0.7 | 0.8 | 0.9 | 1.0 |
|---|---|---|---|---|---|---|---|---|---|---|
| Softmax | **80.8** | **87.1** | **94.0** | **96.6** | **98.2** | **98.8** | **98.8** | **99.1** | **99.2** | **99.1** |
| Linear-norm | 64.8 | 73.6 | 85.7 | 93.2 | 96.6 | 97.5 | 98.2 | 98.8 | 99.0 | 99.1 |
| Linear-weighted | 59.4 | 66.8 | 77.5 | 86.2 | 92.4 | 95.0 | 96.6 | 98.4 | 98.6 | 99.1 |

Table 3: Comprehensiveness (patch-top) results for different MLP attribution methodsn.

| Recover-Top (%) | 0.1 | 0.2 | 0.3 | 0.4 | 0.5 | 0.6 | 0.7 | 0.8 | 0.9 | 1.0 |
|---|---|---|---|---|---|---|---|---|---|---|
| Softmax | **97.8** | **96.9** | **93.9** | **89.9** | **81.6** | **75.5** | **68.9** | **56.8** | **43.5** | **0.0** |
| Linear-norm | 98.3 | 97.1 | 94.8 | 92.3 | 87.7 | 83.5 | 78.2 | 68.8 | 54.9 | 0.0 |
| Linear-weighted | 98.3 | 97.7 | 96.6 | 95.3 | 92.3 | 90.4 | 86.7 | 80.7 | 71.6 | 0.0 |

Table 4: Sufficiency (recover-top) results for different MLP attribution methods.

Overall, the softmax-based attribution consistently outperforms the alternatives across both token removal strategies, supporting its empirical effectiveness despite its heuristic origin.

# B More Results and Experiment Details on Token-Level Output Attribution

In this appendix section, we provide additional experimental details and results for token-level output attribution using DePass across multiple model variants. Our goal is to assess the **faithfulness** of token attributions, measured in terms of **comprehensiveness** and **sufficiency**, following established evaluation metrics.

## B.1 Tasks and Dataset Details

We evaluate our method on two widely used benchmarks that require distinct types of reasoning:

**Known_1000** The Known_1000 dataset [4] consists of factual question-answering prompts, where each prompt targets a known fact (e.g., "Audible.com is owned by Amazon"). This dataset is designed to probe how factual information is stored and retrieved in the model.

**IOI (Indirect Object Identification)** The IOI task [19] involves syntactic reasoning and coreference resolution. Each instance includes a sentence involving two named entities and a pronoun (e.g., "Eleanor and Deanna were thinking about going to the mountain. Eleanor wanted to give a watermelon to →"). The model must correctly resolve the pronoun to the indirect object. This task is well-suited for analyzing compositional reasoning capabilities and token interactions.

## B.2 Faithfulness Evaluation on More Models

We assess our method by comparing its comprehensiveness and sufficiency scores against standard baselines. Comprehensiveness measures how much the model's confidence drops when top-attributed

tokens are removed, while sufficiency assesses how much confidence is retained when only the top-attributed tokens are kept.

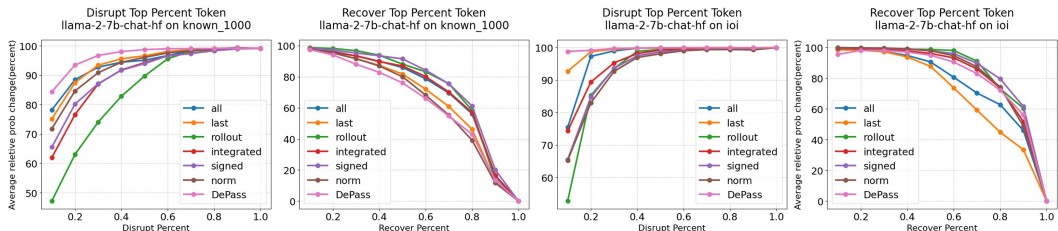

Figure 5: Faithfulness evaluation using Llama-2-7b-chat-hf. DePass achieves a sharper drop in comprehensiveness and higher sufficiency retention, indicating better identification of faithful tokens.

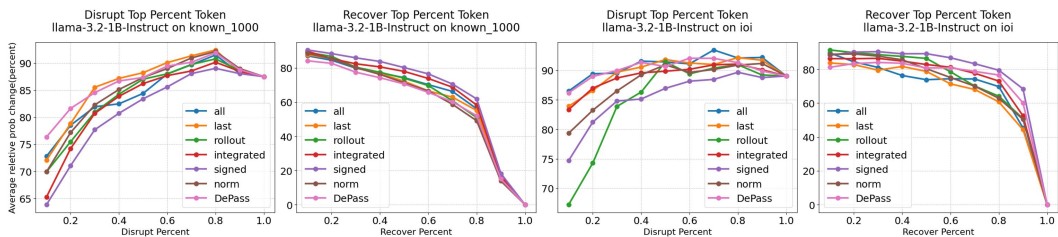

Figure 6: Faithfulness evaluation on Llama-3.2-1B-Instruct. Despite its small scale, DePass effectively captures key tokens contributing to the output, validating its robustness across model sizes.

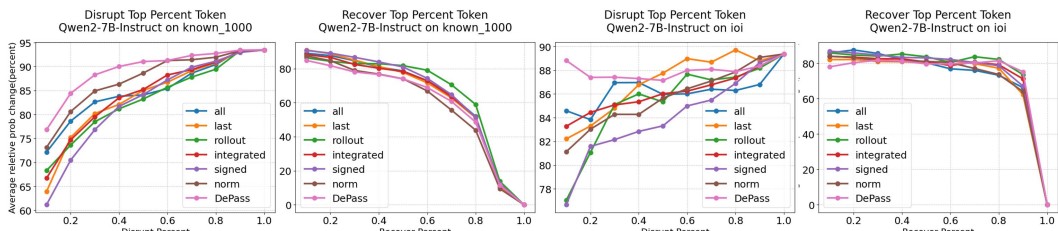

Figure 7: Evaluation on Qwen-2-7B-Instruct across both tasks. DePass consistently outperforms other attribution techniques in both comprehensiveness and sufficiency, demonstrating generalizability across architectures.

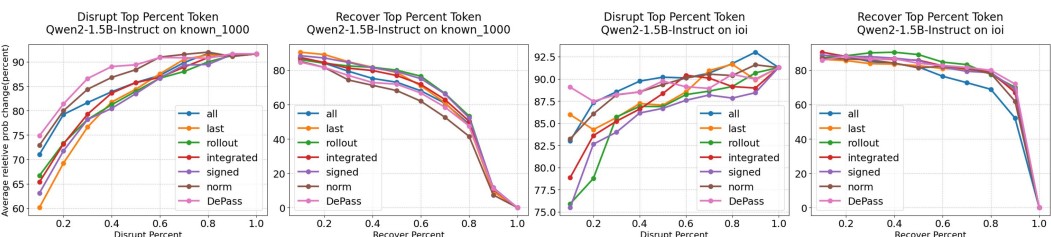

Figure 8: Evaluation on Qwen-2-1.5B-Instruct across both tasks.

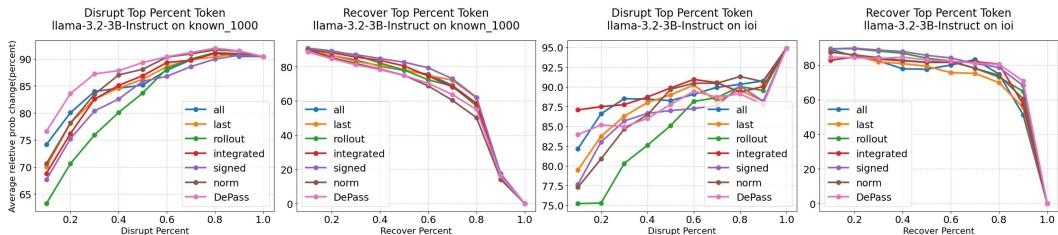

Figure 9: Evaluation on Llama-3.2-3B-Instruct across both tasks.

These results reinforce the effectiveness of DePass in identifying faithfully contributing input tokens across a diverse set of models and tasks.

### B.3 More Cases on Output Attribution

We present additional examples illustrating how DePass assigns output-dependent importance scores to input tokens. These cases highlight DePass's ability to differentiate which parts of the input are most responsible for different model outputs, even under sampling variability.

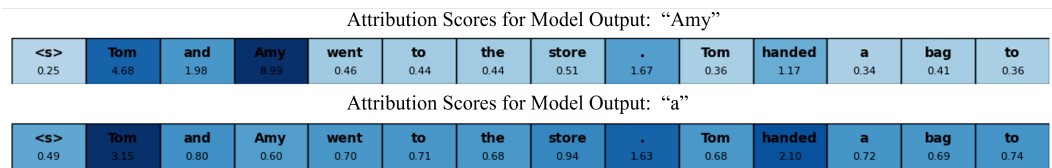

Figure 10: Token-wise output attribution scores from DePass for the prompt *"Tom and Amy went to the store. Tom handed a bag to"*. For two sampled outputs—*"Amy"* and *"a"*—DePass produces distinct attribution patterns, correctly identifying which input tokens support each specific continuation.

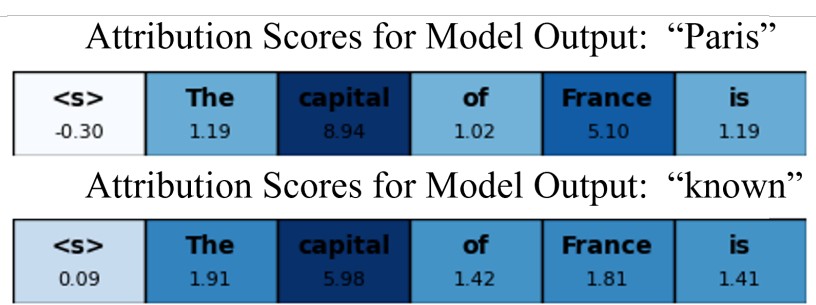

Figure 11: Token-wise output attribution scores for the prompt *"The capital of France is"*. When the model produces *"Paris"* versus *"known"* as outputs, DePass assigns higher relevance to different parts of the prompt accordingly. This shows DePass's sensitivity to output semantics in attributing input importance.

**Illustration for Multi-Token Words' Attribution**   We provide two examples to illustrate how subword tokenization affects DePass scores. For word-level scores, corresponding subword tokens are treated as a **single component from the start of the DePass process**, rather than a simple sum of individual scores.

**Example 1:**

| Token |  | Catal | onia | belongs | to | the | continent | of |
|---|---|---|---|---|---|---|---|---|
| Score | 0.28 | 6.16 | 1.27 | 1.75 | 1.16 | 1.15 | 13.06 | 1.36 |
| Word |  | Catalonia | belongs | to | the | continent | of | |
| Score | 0.28 | 6.38 | 1.92 | 1.37 | 1.38 | 13.24 | 1.59 | |

**Example 2:**

| Token |  | The | mother | tongue | of | Daniel | le | Dar | rie | ux | is |
|---|---|---|---|---|---|---|---|---|---|---|---|
| Score | 0.30 | 0.87 | 2.09 | 9.69 | 1.23 | -0.01 | 1.31 | 2.00 | 2.69 | 1.36 | 0.73 |
| Word |  | The | mother | tongue | of | Danielle | Darrieus | is | | | |
| Score | 0.30 | 1.12 | 2.29 | 9.85 | 1.52 | 1.85 | 4.12 | 1.18 | | | |

Both token-level and word-level attributions with DePass are effective. While token-level scores may disperse across subword tokens, they remain highly informative for distinguishing important tokens. Higher scores on specific subwords can even indicate triggers for the model's memory of the full word. DePass naturally supports aggregating subword tokens for desired word-level attribution.

**Token-Level Attribution Examples for Multiple Attribution Methods**  The following tables present token-level attribution scores computed by DePass and several baseline methods for illustrative prompts. Each table corresponds to a different target token for the same input, highlighting how DePass distributes relevance across input tokens in a context-sensitive manner. Raw DePass scores are provided alongside normalized values to facilitate comparison with other attribution methods.

| Token | DePass | Normalized DePass | All | Last | Rollout | Integrated Gradients | Signed | Norm |
|---|---|---|---|---|---|---|---|---|
|  | -0.30 | -0.02 | 0.30 | 0.28 | 0.35 | 0.00 | 0.17 | 0.40 |
| The | 1.20 | 0.07 | 0.14 | 0.14 | 0.13 | 0.01 | 0.17 | 0.14 |
| capital | 8.94 | 0.52 | 0.14 | 0.14 | 0.13 | 0.84 | 0.16 | 0.17 |
| of | 1.02 | 0.06 | 0.14 | 0.14 | 0.13 | 0.14 | 0.17 | 0.14 |
| France | 5.06 | 0.30 | 0.14 | 0.15 | 0.13 | 0.00 | 0.16 | 0.07 |
| is | 1.20 | 0.07 | 0.14 | 0.15 | 0.13 | 0.01 | 0.17 | 0.09 |

Table 5: Token-level attribution scores for the prompt "The capital of France is" with target "Paris".

| Token | DePass | Normalized DePass | All | Last | Rollout | Integrated Gradients | Signed | Norm |
|---|---|---|---|---|---|---|---|---|
|  | 0.09 | 0.01 | 0.30 | 0.28 | 0.35 | 0.26 | 0.18 | 1.00 |
| The | 1.91 | 0.15 | 0.14 | 0.14 | 0.13 | 0.00 | 0.08 | 0.00 |
| capital | 6.00 | 0.48 | 0.14 | 0.14 | 0.13 | 0.65 | 0.20 | 0.00 |
| of | 1.41 | 0.11 | 0.14 | 0.14 | 0.13 | 0.00 | 0.16 | 0.00 |
| France | 1.80 | 0.14 | 0.14 | 0.15 | 0.13 | 0.00 | 0.22 | 0.00 |
| is | 1.41 | 0.11 | 0.14 | 0.15 | 0.13 | 0.09 | 0.17 | 0.00 |

Table 6: Token-level attribution scores for the prompt "The capital of France is" with target "known".

| Token | DePass | Normalized DePass | All | Last | Rollout | Integrated Gradients | Signed | Norm |
|-------|--------|-------------------|-----|------|---------|----------------------|--------|------|
|  | 0.32 | 0.02 | 0.14 | 0.12 | 0.17 | 0.02 | 0.07 | 0.88 |
| gra | 1.52 | 0.11 | 0.07 | 0.07 | 0.06 | 0.00 | 0.06 | 0.00 |
| pe | 0.08 | 0.01 | 0.07 | 0.07 | 0.06 | 0.00 | 0.10 | 0.00 |
| : | 0.20 | 0.01 | 0.07 | 0.07 | 0.06 | 0.00 | 0.06 | 0.11 |
| pur | 5.25 | 0.36 | 0.07 | 0.07 | 0.06 | 0.00 | 0.07 | 0.00 |
| ple | 0.14 | 0.01 | 0.07 | 0.07 | 0.06 | 0.00 | 0.08 | 0.00 |
| , | 0.10 | 0.01 | 0.07 | 0.07 | 0.06 | 0.00 | 0.06 | 0.00 |
| ban | 0.48 | 0.03 | 0.07 | 0.07 | 0.06 | 0.00 | 0.08 | 0.00 |
| ana | 0.12 | 0.01 | 0.07 | 0.07 | 0.06 | 0.00 | 0.05 | 0.00 |
| : | 0.15 | 0.01 | 0.07 | 0.07 | 0.06 | 0.98 | 0.09 | 0.00 |
| yellow | 1.53 | 0.11 | 0.07 | 0.07 | 0.06 | 0.00 | 0.05 | 0.00 |
| , | 0.19 | 0.01 | 0.07 | 0.07 | 0.06 | 0.00 | 0.08 | 0.00 |
| apple | 4.16 | 0.29 | 0.07 | 0.07 | 0.06 | 0.00 | 0.08 | 0.00 |
| : | 0.17 | 0.01 | 0.07 | 0.07 | 0.06 | 0.00 | 0.06 | 0.00 |

Table 7: Token-level attribution scores for fruit-color completion task with target "red".

DePass reports both raw and normalized attribution scores, with raw values summing exactly to the output logit. This allows direct interpretation of each score's impact on the prediction. Attribution distributions vary across different targets for the same input, reflecting the model's context-dependent reasoning. DePass consistently assigns attribution to semantically meaningful input tokens, whereas baseline and gradient-based methods often fail to highlight relevant tokens or exhibit clear semantic alignment.

# C   More Results and Experiment Details on Token-Wise Subspace Attribution

## C.1   Prompt Construction

### C.1.1   CounterFact Prompt Construction

The **CounterFact** dataset [4] is designed to evaluate factual recall in language models. Each instance provides:

- a `subject` entity (e.g., `Go Hyeon-jeong`),
- a `target` (the correct answer, e.g., `Korean`),
- a `target_new` (an incorrect, but plausible alternative, e.g., `French`),
- multiple natural language `prompts` querying the relation (e.g., "The mother tongue of Go Hyeon-jeong is").

We construct multiple prompt variants per example to simulate different reasoning settings. Specifically, we create three types of prompts:

- **Initialization Prompt (No Information)**: A question-only prompt constructed by sampling one of the available templates, without providing supporting information.
- **Untruthful Prompt**: A misleading information context is constructed by inserting the subject along with the incorrect fact (`target_new`) into a randomly selected prompt template from the provided list. This prompt is then used as external information preceding the same question template, simulating a factual error in the input.

Each final prompt is formatted as follows:

```
According to the given information and your knowledge, answer
the question.
Information:  <Inserted Factual or Counterfactual Sentence>
(optional)
Question:  <Sampled Question Template>
The answer is:
```

During construction, we ensure diversity by randomly sampling prompt templates for both the question and the information. For each data point, one question prompt is selected and paired with either the correct or incorrect fact from the candidate list.

### C.1.2 TruthfulQA Prompt Construction

The **TruthfulQA** dataset [23] contains questions designed to probe language models' susceptibility to producing factually incorrect or misleading answers. For each question, it provides a set of factually correct answers (`mc1_targets` and `mc2_targets`) and a pool of plausible but incorrect ones.

We construct two types of prompts per example:

- **Initialization Prompt (No Information)**: A question-only multiple-choice prompt where the model must choose the best answer from a randomized mix of one correct and one incorrect option, without access to any supporting context.

- **Untruthful Prompt**: A misleading information-bearing prompt, where a randomly sampled incorrect statement from the provided `mc1_targets` or `mc2_targets` is inserted as `Information`. The model must then answer the same question with the same randomized options, now influenced by the incorrect prior.

Each prompt is formatted as follows:

```
According to the given information and your knowledge, choose
the best choice from the following options.
Information:  <Random Incorrect Statement> (optional)
Question:  <Original Question>
Options:  A: <Incorrect Option>   B: <Correct Option> (random
order)
The answer is:
```

To ensure variation and avoid position bias, we shuffle the order of the correct and incorrect choices. In all cases, the model is required to pick between exactly two answer candidates, enabling us to isolate the effect of misleading information on model predictions.

### C.2 Truthful Probe Training and Evaluation

**Probe Training**   Following [22], we train linear probes on hidden states from transformer-based language models to detect factual consistency. We use the **CounterFact** and **TruthfulQA** datasets, each labeled with truthful and untruthful prompts. The datasets are split into training and testing sets with balanced labels to ensure fair evaluation.

For each prompt, we extract the hidden states of the final token across all transformer layers. These per-layer activations serve as features for training logistic regression classifiers (one per layer) to distinguish between truthful and untruthful inputs. We use the `saga` solver in `scikit-learn`, with a learning rate of $0.01$ and maximum iteration count of $1000$.

During training, we shuffle and concatenate the hidden states of truthful and untruthful samples and fit each layer-specific classifier independently. After training, we evaluate the classifiers on a held-out test set to assess the layer-wise linear separability of factual information.

**Probe Evaluation**   Figure 12 and Figure 14 show the classification accuracy for different Llama and Qwen model variants. We observe a consistent trend across models: classifier accuracy improves with depth, often exceeding 90% in the middle-to-late layers. This suggests that factual signals become increasingly linearly separable in deeper representations.

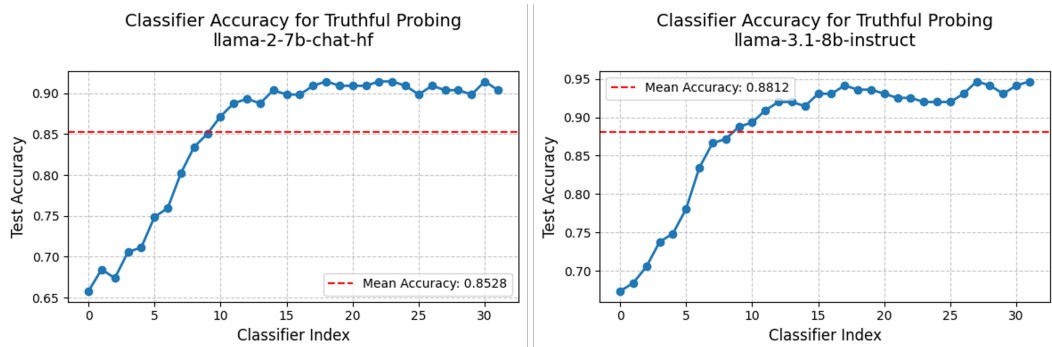

Figure 12: Truthful classifier accuracy on Llama-2-7B-chat-hf and Llama-3.1-8B-Instruct.

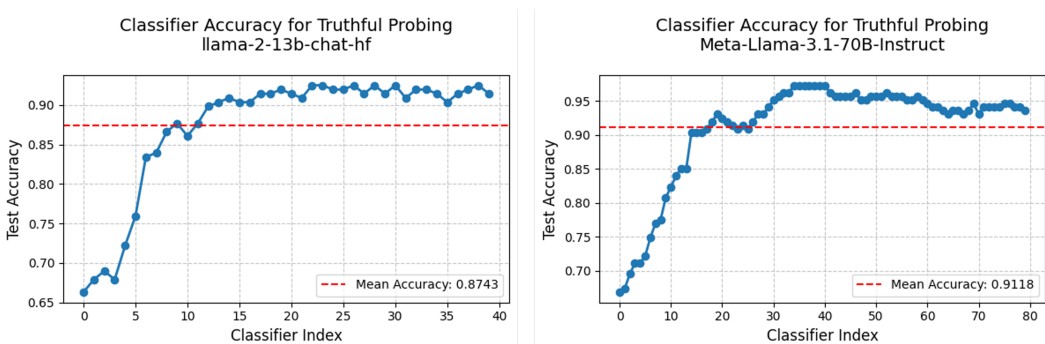

Figure 13: Truthful classifier accuracy on Llama-2-13b-chat-hf and Meta-Llama-3.1-70B-Instruct

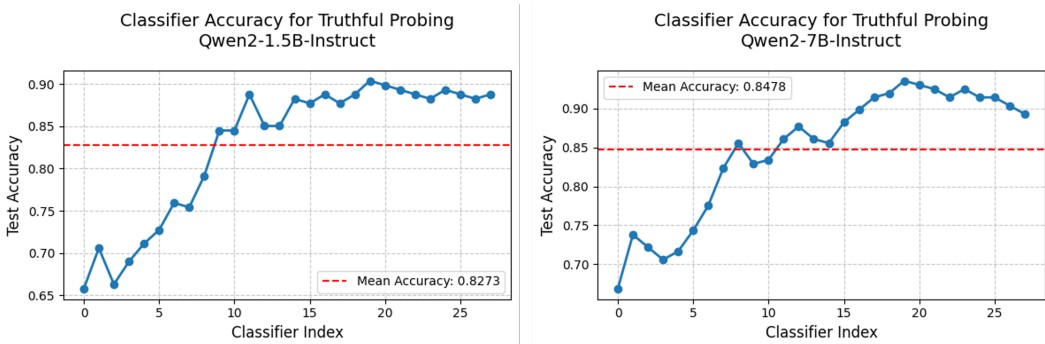

Figure 14: Truthful classifier accuracy on Qwen-2-1.5B-Instruct and Qwen-2-7B-Instruct.

### C.3  Masking Strategy

**Token selection for masking.** For a given prompt, we extract the hidden states of all tokens across all layers. Based on classifier accuracy trends, we consider only classifiers from layer 10 onward, where truthfulness signals are most separable. For each token, we compute its average predicted probability of being *untruthful* across these layers.

Tokens with a mean probability $\leq 0.5$ are retained. For those with probability $> 0.5$, we apply two masking strategies:

1. **Direct masking (TACS)[22]:** Remove all tokens predicted as untruthful ($p > 0.5$) from the input and re-run the model.
2. **DePass-based masking:** For each token identified as untruthful, we decompose its hidden state to attribute contributions from other input tokens. We average the untruthful tokens'

contribution vectors to compute a global *untruthfulness attribution score*. Then, we identify the top $k$ contributing tokens in the input (matching the number of untruthful tokens in direct masking) and remove them.

These complementary methods help evaluate whether selectively removing suspected untruthful information can steer model predictions towards higher factuality, while controlling for deletion magnitude.

### C.4  More Cases on Subspace Attribution

We present additional untruthful examples to further illustrate how DePass attributes subspace-level information to input tokens.

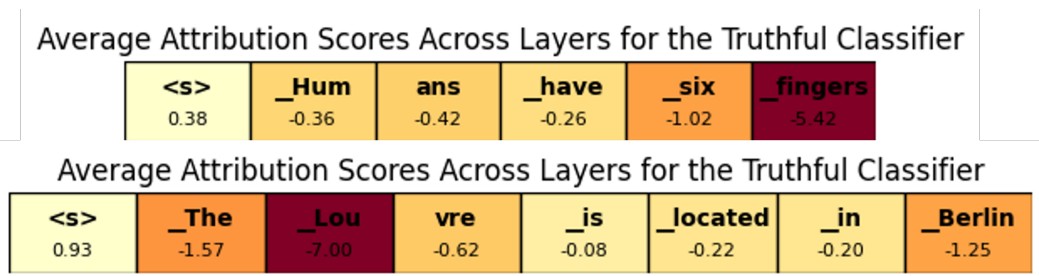

Figure 15: Additional examples demonstrating DePass's ability to attribute information from specific subspaces back to input tokens. Attribution scores are computed with respect to the output label 0 (untruthful); more negative scores indicate stronger contributions toward untruthful signal.

### C.5  Case Studies: Potential of Combining DePass with SAE

**Combining DePass with SAE.** While SAEs can uncover meaningful features, they do not inherently indicate which input tokens activate those features. DePass provides a complementary capability: by decomposing activations into additive components, it flexibly connects different elements of a model—tokens, components, and subspaces—with SAE features. This enables fine-grained analysis of how interpretable features are triggered by specific tokens and how they propagate through the model. Below, we present several case studies demonstrating the potential of combining DePass with SAE for token-level attribution. The feature annotations used in these examples are drawn from Neuronpedia[4], and all cases are based on LLaMA-3.1-8B with the corresponding SAE [45].

**Case 1: Climate Change.**  *Prompt:* "In exploring the relationship between climate change and urban planning, the research identifies key strategies for sustainable development. It underscores the urgency of integrating environmental considerations into city planning."

*Feature:* Layer 31, Feature 5226 – "climate change and its associated impacts"

We analyze three tokens that activate this feature. DePass identifies that the activation of the SAE feature is primarily driven by the semantic contribution of `climate` (6), which then propagates to other relevant tokens such as `change` (7) and `environmental` (27) (Table 8).

---

[4]https://www.neuronpedia.org/

| Token analyzed (index) | Activation value | Source token (index) | Contribution |
| --- | --- | --- | --- |
| **climate (6)** | 35.2500 | climate (6) | 26.8750 |
| | | In (1) | 3.0781 |
| | | exploring (2) | 1.8750 |
| | | the (3) | 1.4844 |
| | | relationship (4) | 1.4844 |
| | | between (5) | 1.4844 |
| **change (7)** | 15.6875 | climate (6) | 16.5000 |
| | | relationship (4) | 0.8594 |
| | | between (5) | 0.3984 |
| | | change (7) | 0.5586 |
| | | In (1) | -1.9141 |
| **environmental (27)** | 7.4688 | climate (6) | 9.4375 |
| | | exploring (2) | 1.0234 |
| | | environmental (27) | -0.6562 |
| | | underscores (22) | -0.1670 |
| | | urgency (24) | -0.2432 |
| | | integrating (26) | -0.3516 |

Table 8: DePass token-level attribution of SAE feature 5226 ("climate change and its associated impacts").

**Case 2: Simplicity in Context.** *Prompt:* "Avocado toast, a simple yet trendy breakfast option, gains a delightful twist with a sprinkle of chili flakes and a drizzle of honey for that perfect balance of spicy and sweet."

*Feature:* Layer 31, Feature 26874 – "the context of simplicity in various contexts"

DePass shows that the activation of this feature is largely driven by the semantic link from `simple` `(6)` to tokens such as `yet` `(7)` and punctuation, highlighting how contextual modifiers contribute to SAE features (Table 9).

| Token analyzed (index) | Activation value | Source token (index) | Contribution |
| --- | --- | --- | --- |
| **yet (7)** | 7.4062 | simple (6) | 5.8438 |
| | | toast (3) | 1.9219 |
| | | yet (7) | 1.1953 |
| | | , (4) | -0.5312 |
| | | a (5) | -0.5898 |
| | | `<bos>` (0) | -1.3750 |
| **,(11)** | 3.5312 | simple (6) | 1.7031 |
| | | toast (3) | 0.8125 |
| | | yet (7) | 0.7344 |
| | | `<bos>` (0) | -1.8906 |

Table 9: DePass token-level attribution of SAE feature 26874 ("simplicity in context").

**Case 3: Economic Concepts.** *Prompt:* "In economics, supply and demand is a model for understanding how prices and quantities are determined in a market system. This concept is foundational in economic theory and affects various market structures."

*Feature:* Layer 25, Feature 9618 – "references to economic topics and concepts"

Here, DePass reveals that the activation of the "economic concepts" feature stems from semantic contributions of `economics` `(2)` that propagate to `economic` `(28)` and `and` `(30)` (Table 10).

Across these examples, combining DePass with SAE enables precise token-level attribution of interpretable features. Whereas SAE identifies semantically coherent features, DePass traces their origins and propagation across tokens, yielding a more complete mechanistic account of model representations.

| Token analyzed (index) | Activation value | Source token (index) | Contribution |
|---|---|---|---|
| **economic (28)** | 15.0625 | economics (2) | 5.9375 |
| | | economic (28) | 4.9062 |
| | | <bos> (0) | 0.9805 |
| | | . (22) | 0.1348 |
| **and (30)** | 7.3438 | economics (2) | 4.7500 |
| | | economic (28) | 1.9844 |
| | | <bos> (0) | 1.1719 |
| | | concept (24) | -0.1025 |
| | | In (1) | -0.6680 |

Table 10: DePass token-level attribution of SAE feature 9618 ("economic topics and concepts").

# D    More Results and Experiment Details on Component-Wise DePass

## D.1    Additional Model Component-wise Attribution Results on Different Models

To further evaluate the generality and robustness of DePass across architectures and scales, we present additional results on component-wise attribution. Specifically, we assess the ability of DePass to identify important **attention heads** and **MLP neurons** across various model families, including LLaMA-3.2 and Qwen2 at both small and medium scales.

We apply Top-$k$ and Bottom-$k$ masking strategies: masking the top-$k$ most attributed components should lead to a notable performance drop, while masking the bottom-$k$ should minimally impact output quality. Across all settings, DePass demonstrates strong alignment between attribution scores and functional importance.

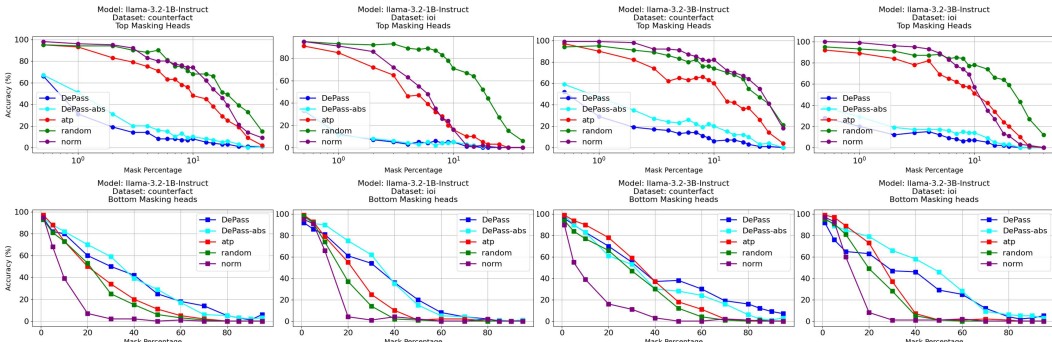

Figure 16: Top-$k$ and Bottom-$k$ masking results for **attention head** attribution on Llama-3.2-1B-Instruct and Llama-3.2-3B-Instruct. DePass leads to a larger accuracy drop with Top-$k$ masking and better retention with Bottom-$k$ masking, highlighting its effectiveness in identifying key attention heads across different model scales.

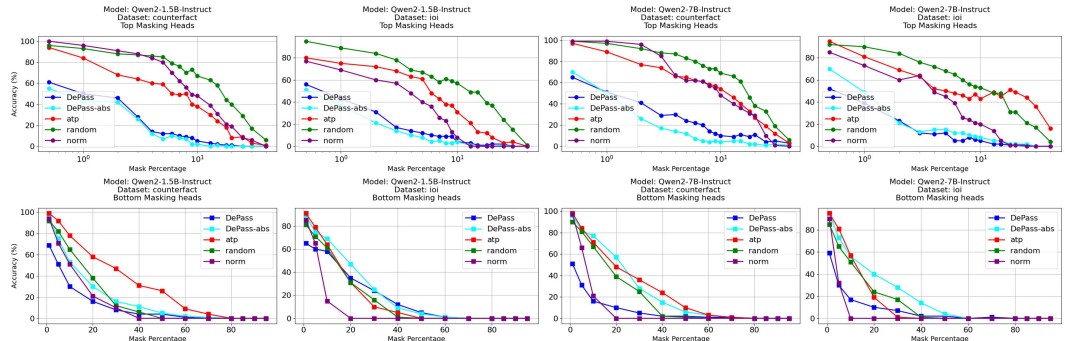

Figure 17: Top-$k$ and Bottom-$k$ masking evaluation on Qwen2-1.5B-Instruct and Qwen2-7B-Instruct for **attention head** attribution. DePass consistently achieves greater accuracy drop under Top-$k$ masking and stronger performance retention under Bottom-$k$ masking, demonstrating its ability to pinpoint influential attention heads across architectures.

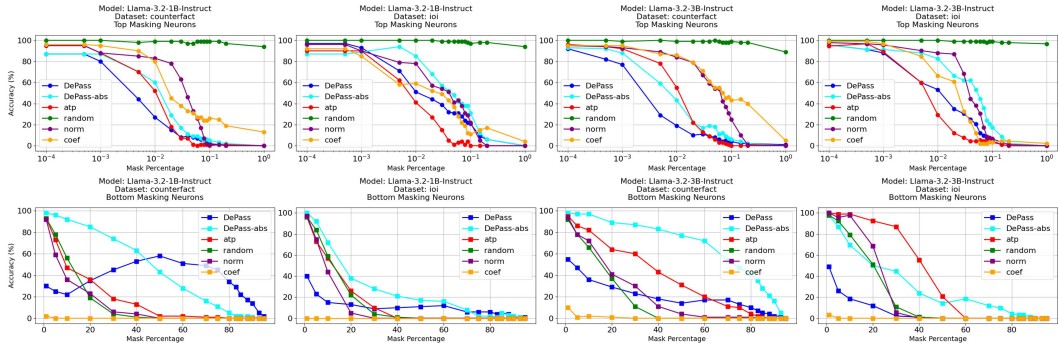

Figure 18: Top-$k$ and Bottom-$k$ masking evaluation on Llama-3.2-1B-Instruct and Llama-3.2-3B-Instruct for **MLP neuron** attribution. DePass outperforms baselines by inducing sharper performance degradation when top neurons are masked and maintaining better accuracy when only low-ranked neurons are ablated.

## D.2 Efficiency of Neuron-level Attribution

To ensure the practicality of fine-grained attribution, we compare the runtime of DePass with the standard ablation approach when attributing neurons in intermediate MLP layers of different LLaMA models. As shown in Table 11, DePass provides a significant speedup, reducing computation time by up to two orders of magnitude, while maintaining attribution fidelity. This efficiency makes DePass especially suitable for scaling to larger models and longer prompts.

Table 11: Runtime comparison for neuron-level attributions over an intermediate MLP layer. DePass achieves substantial acceleration over the ablation baseline.

| Model | Intermediate Size | Method | Time (s) |
|---|---|---|---|
| LLaMA-2-7B-Chat | 11008 | Ablation | 321.04 |
| | | DePass | **7.22** |
| LLaMA-3.2-3B-Instruct | 8192 | Ablation | 234.77 |
| | | DePass | **2.91** |
| LLaMA-3.2-1B-Instruct | 8192 | Ablation | 134.76 |
| | | DePass | **2.23** |

## D.3 Important Head and Neuron Distributions

We provide a detailed analysis of the internal model components by visualizing the distributions of top-attributed **attention heads** and **MLP neurons**, both at the dataset level and for individual prompts.

These analyses reveal how different models and attribution methods identify key substructures that contribute to model predictions.

### D.3.1 Average Distribution over the Dataset

To better understand which internal components contribute most to model predictions, we visualize the average importance scores assigned by **DePass** to attention heads and MLP neurons across two representative datasets: IOI and CounterFact. These aggregated views highlight consistent attribution patterns across layers and architectures.

For attention heads, each heatmap aggregates importance scores across all prompts, with rows corresponding to transformer layers and columns representing head indices. For MLP neurons, we group neurons into bins of 100 for ease of visualization, plotting the average importance per bin across datasets.

These visualizations provide structural insights into how different model components are utilized in practice, and where important predictive capacity is concentrated across architectures and tasks.

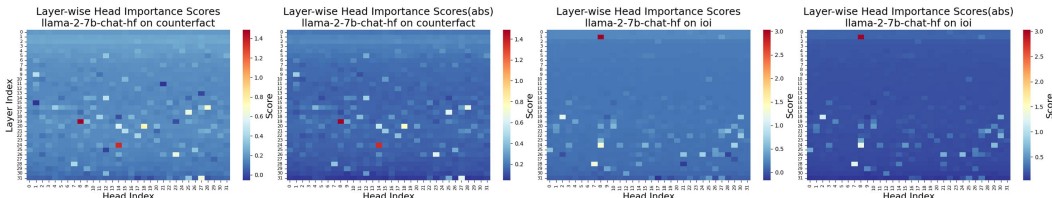

Figure 19: Average attention head importance identified by DePass on Llama-2-7b-chat-hf, aggregated across datasets. Rows indicate transformer layers; columns indicate head indices.

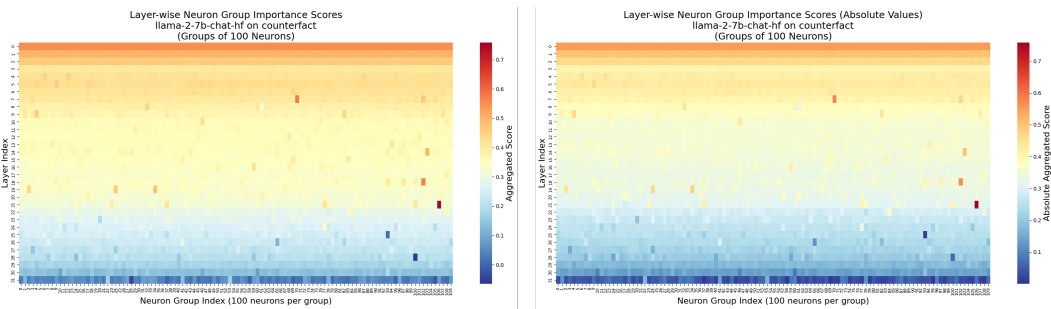

Figure 20: Average MLP neuron importance identified by DePass on Llama-2-7b-chat-hf, aggregated across datasets. Neurons are grouped in bins of 100 for visualization.

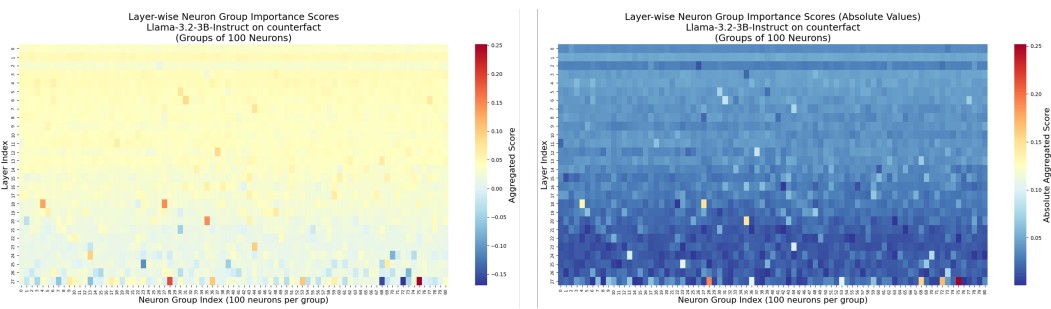

Figure 21: Average MLP neuron importance identified by DePass on Llama-3.2-3B-Instruct, aggregated across datasets. Neurons are grouped in bins of 100 for visualization.

### D.3.2 Per-Prompt Case Analysis

We further present case-by-case visualizations to analyze the distribution of important attention heads and MLP neurons for individual prompts. These visualizations compare **DePass** with baseline attribution methods (e.g., AtP, Norm) and aim to identify the components most responsible for producing the correct answer under each input. In contrast to dataset-level views, these examples highlight how attribution patterns vary across prompts and methods.

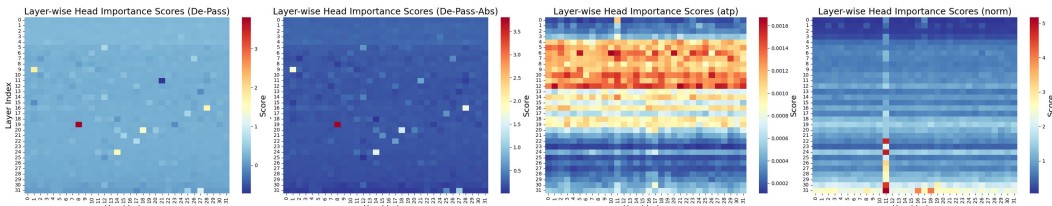

Figure 22: Attention head attribution for the prompt ``The mother tongue of Danielle Darrieux is'', ``French'' using Llama-2-7b-chat-hf. DePass more effectively isolates attention heads that contribute to generating the correct answer compared to baselines.

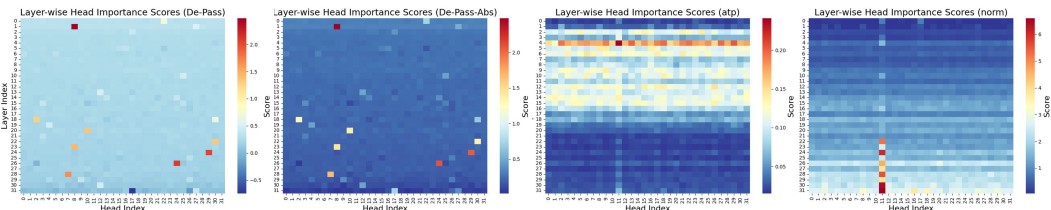

Figure 23: Attention head attribution for the prompt ``After the lunch, Mildred and Eleanor went to the market.  Eleanor gave a watermelon to'', ``Mildred'' using Llama-2-7b-chat-hf. DePass reveals a more focused and task-relevant distribution of important attention heads.

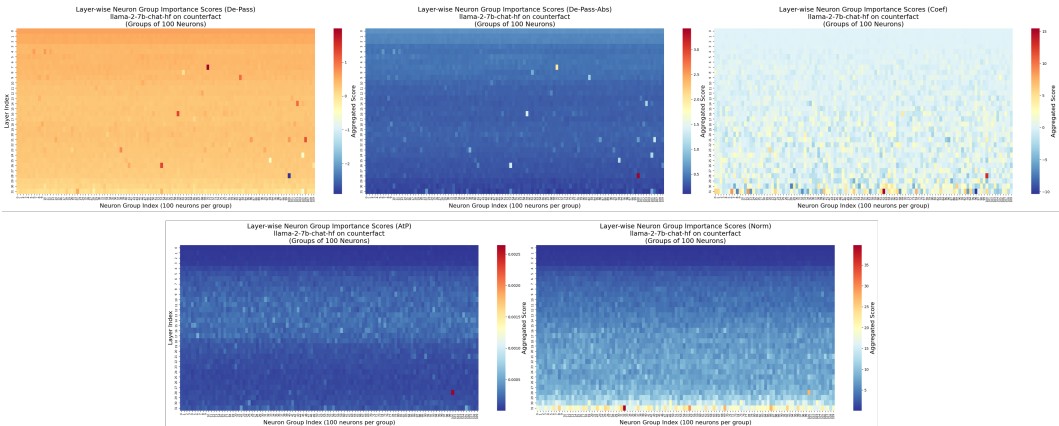

Figure 24: MLP neuron attribution for the prompt ``The mother tongue of Danielle Darrieux is'', ``French'' using Llama-2-7b-chat-hf. Neurons are grouped into bins of 100 for visualization. DePass effectively identifies neurons critical for predicting the correct answer.

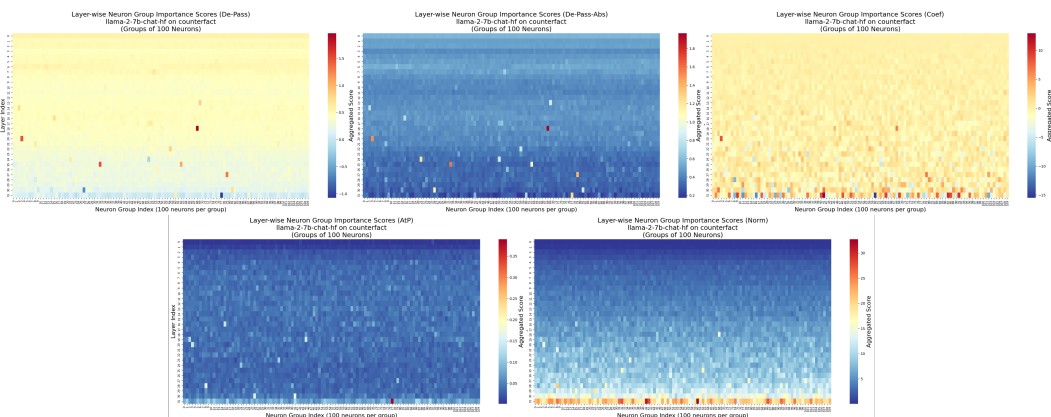

Figure 25: MLP neuron attribution for the prompt ``Tarvaris Jackson plays in the position of'', ``quarterback'' using Llama-2-7b-chat-hf. Neurons are grouped into bins of 100 for visualization. DePass more accurately isolates neurons essential for producing the correct answer compared to baselines.

## E  More Results and Experiment Details on Subspace-Wise DePass

### E.1  Classifier Training and Evaluation

**Dataset for Probing Classifier Training**   To construct a multilingual dataset with identical semantics across different languages, we translate a subset of the CounterFact[4] dataset into multiple languages while ensuring that the meaning remains strictly unchanged. This allows us to probe whether the identified important subspaces generalize across linguistic variations that share the same semantic content. An example entry is shown below:

> **Original (English):**
> "prompt":  "The mother tongue of Danielle Darrieux is",
> "answer":  "French"
>
> **French:**
> "prompt":  "La langue maternelle de Danielle Darrieux est le",
> "answer":  "français"
>
> **German:**
> "prompt":  "Die Muttersprache von Danielle Darrieux ist",
> "answer":  "Französisch"
>
> **Italian:**
> "prompt":  "La lingua madre di Danielle Darrieux è il",
> "answer":  "francese"

**Classifier Training Details**   We then use this multilingual, semantically aligned dataset to train a language classifier. This setup allows us to evaluate whether the subspace directions identified by DePass are stable and predictive across languages, thus offering a rigorous test of attribution generalization.

For each input prompt, we extract the hidden state of the final token and use it as the input feature to the classifier. The corresponding language of the prompt serves as the class label. We train a separate multi-class classifier at each transformer layer using the translated subset of the CounterFact dataset, which is evenly split into training and testing sets with balanced label distributions across languages.

We use logistic regression implemented via the `scikit-learn` library, with the `saga` solver, a learning rate of $0.01$, and a maximum iteration count of $1000$. During training, we concatenate the hidden states across all prompts and shuffle the dataset to ensure robustness and reduce bias. This approach enables us to analyze which layers encode language-distinguishing information and how DePass-selected subspaces contribute to that encoding.

**Classifier Evaluation**  The language classifier achieves high accuracy across both models, indicating that language-specific directions are reliably encoded in the hidden states.

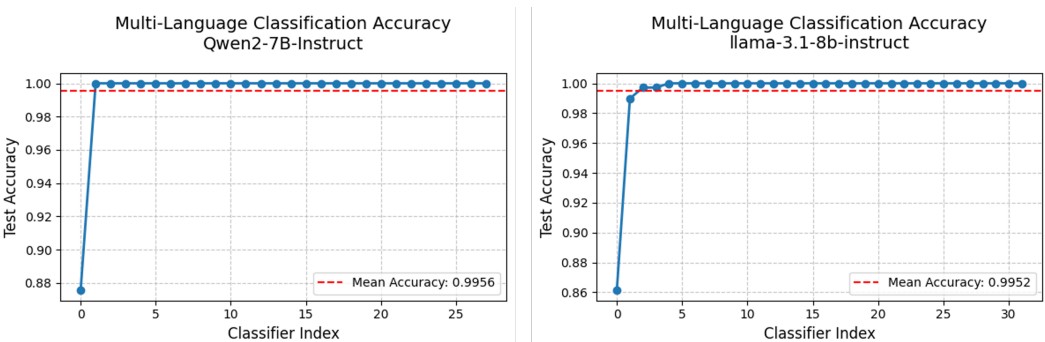

Figure 26: Accuracy of the language classifier across different layers on Llama-3.1-8B-Instruct and Qwen2-7B-Instruct. Results show strong separability of language-specific information in hidden representations.

## E.2   Subspace Projection

We describe how to construct the projection matrix $\mathbf{P}_t \in \mathbb{R}^{d \times d}$ used to decompose hidden states into components within and orthogonal to a target subspace. This subspace can be defined by any set of directions of interest, such as the row space of a linear classifier, neuron activations, or gradient-based attribution vectors.

Given a matrix $W \in \mathbb{R}^{d \times c}$ whose column space spans the target subspace (e.g., the weight matrix of a linear classifier), we compute the singular value decomposition (SVD) of $W^\top$ as:

$$W^\top = U \Sigma V^\top,$$

and retain the top-$r$ left singular vectors in $U$, where $r = \operatorname{rank}(W)$. These vectors form an orthonormal basis $U_r \in \mathbb{R}^{d \times r}$ for the subspace of interest.

The projection matrix onto this subspace is given by:

$$\mathbf{P}_t = U_r U_r^\top,$$

and its orthogonal complement is $\mathbf{I} - \mathbf{P}_t$, where $\mathbf{I}$ is the identity matrix.

For any hidden state $x_i \in \mathbb{R}^D$, we obtain the decomposition:

$$x_i^\| = \mathbf{P}_t x_i, \quad x_i^\perp = (\mathbf{I} - \mathbf{P}_t) x_i,$$

where $x_i^\|$ lies in the target subspace and $x_i^\perp$ is orthogonal to it. This decomposition enables precise attribution and intervention by isolating the contribution of specific representational directions to model behavior.

## E.3   More Results on Qwen2-7B-Instruct

We present further analysis demonstrating the subspace attribution capability of DePass on Qwen2-7B-Instruct. To validate the separation of representational content, we apply t-SNE to the language subspace projections $X_{\text{lang}}^{\text{dec}}$ obtained from multilingual inputs. As shown in Figure 27, the resulting embeddings form distinct clusters based on language identity. This indicates that DePass successfully isolates language-specific patterns within a dedicated subspace, supporting its ability to disentangle and attribute form-related features.

We further examine subspace semantics by decoding from both the `language` and `semantic` subspaces for a shared English input: *"In which city did Charles Alfred Pillsbury's life end?"* As shown in Table 12, tokens from the language subspace are dominated by structural and frequent function words characteristic of each language, while tokens from the semantic subspace correspond to factual entities such as names and cities. This further supports that DePass routes distinct representational factors—form and meaning—into appropriate subspaces.

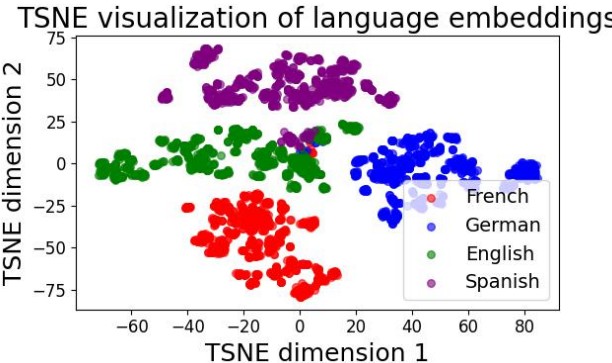

Figure 27: t-SNE visualization of token-wise representations in the `language` subspace ($X_{\text{lang}}^{\text{dec}}$) from multilingual prompts in Qwen2-7B-Instruct. The clear clustering by language confirms that DePass accurately attributes language-specific signals to this subspace.

| Language | Decoded Tokens from Language Subspace | Decoded Tokens from Semantic Subspace |
|---|---|---|
| English | San, ", __, New, _ | order, aug, which, Charles, Charles |
| French | quoi, chaque, cette, son, l | Minneapolis, vec, cause, partir, uc |
| German | und, der, die, San, das | Minneapolis, ámb, Stockholm, Cambridge, St |
| Spanish | las, el, ¿, los, la | Minneapolis, Bloom, Saint, ven, och |

Table 12: Tokens decoded from the `language` and `semantic` subspaces of Qwen2-7B-Instruct. The language subspace captures structural and high-frequency tokens, while the semantic subspace reflects content-specific entities relevant to the input query.

## F  Implementation Details

We run all experiments on a cluster of A6000 GPUs. Since DePass operates entirely via forward decomposition without requiring model fine-tuning or backpropagation, our method is highly efficient and incurs minimal memory overhead. All experiments are conducted in PyTorch, using pre-trained models with lightweight modifications to enable decomposition-aware forward passes.

## G  Limitations of DePass

To precisely trace information flow within the model, DePass freezes the attention scores during decomposed forward pass. As a result, under the Transformer circuits framework [52], DePass captures only the flow through the output-value (OV) circuit, without explaining query-key (QK) circuit interactions. Decomposing the QK circuit remains a challenging open problem: first, QK and VO circuits are entangled yet structurally distinct, making it difficult to construct human-interpretable representations of their interactions; second, following DePass's additive decomposition approach, attribution of QK circuit would involve a quadratic complexity explosion. We believe that unwrapping the QK circuit is a key direction for future work.

DePass is compatible with most Transformer architectures. In this paper, we validate it on two widely used families—LLaMA and Qwen—across different model sizes. Extending DePass to other architectures primarily requires adapting to differences in LayerNorm implementation and updating module names at the code level. We plan to release a general-purpose toolkit supporting a broad range of Transformer models.

While DePass supports one-pass attribution across all model components and, in theory, allows hidden states to be decomposed into an infinite number of parts, practical memory constraints require grouping components and computing their forward passes sequentially. This can lead to increased time costs when a finer-grained analysis is desired. Future work includes optimizing computational efficiency and improving hardware deployment strategies to better support attribution with higher resolution.

