# OpenReview forum: "DePass: Unified Feature Attributing by Simple Decomposed Forward Pass"
_NeurIPS.cc/2025/Conference — NeurIPS 2025 poster_

### Official Review · Reviewer_w5Wf · 2025-06-27

**Clarity:** 3
**Significance:** 4
**Originality:** 3
**Rating:** 4
**Confidence:** 3

**Summary:**

The authors propose DePass, a mechanistic interpretability framework for Transformer decoders, which decomposes the forward pass. This method partitions the hidden states into additive components that are propagated through the model while keeping the attention scores and MLP activations fixed. This decomposition allows for the precise attribution of model outputs to interpretable components at various levels of granularity. DePass does not require gradients or retraining. The paper demonstrates the flexibility and fidelity of the method through experiments on factual question answering and indirect object identification tasks.

**Questions:**

Please refer to weaknesses above.

**Ethical Concerns:**

["NO or VERY MINOR ethics concerns only"]

**Final Justification:**

The authors addressed the questions I had, and I think it is a valuable idea above acceptance threshold. For this reason I mantain my initial score.

**Limitations:**

The trade-off introduced by freezing attention and MLP activations is only briefly mentioned in the appendix and should be discussed in the main text. The computational overhead is not analyzed empirically. Additionally, a brief note on the potential dual-use risks of interpretability tools would be beneficial.

**Quality:**

3

**Strengths And Weaknesses:**

Strength:
The idea of decomposing the forward pass into additive components is an elegant and clearly described contribution. Although conceptually related approaches exist, this work generalizes them to a broader setting.
A major strength is the versatility of DePass across various attribution granularities, showcasing its extensibility which is interesting for the community as a tool.
The authors demonstrate the method across multiple model families, with evaluations on standard attribution benchmarks, and compelling qualitative examples (e.g., language-vs-semantic subspace). The comparisons to gradient- and attention-based attribution baselines are appropriate and favorable.
The paper is well-organized, clearly written, and generally accessible to readers familiar with mechanistic interpretability.

Weaknesses:
The method freezes attention scores and MLP activations across all components, which simplifies decomposition but could break causal faithfulness. This tradeoff is briefly noted in the appendix but should be discussed more thorough discussion in the main paper.
The MLP attribution rule is based on a softmax over pre-activation dot products. While intuitive, this approach is heuristic and lacks theoretical justification or empirical comparison to alternatives. The effect of this design choice on attribution accuracy is not evaluated.
While the method is compared to gradient-based and attention-based baselines, it omits evaluation against more mechanistic or causal interpretability methods such as activation patching, ablation-based attribution, or residual stream tracing. This weakens the claim of offering a foundational framework.
The discussion of computational efficiency is limited. Although the authors argue for parallelizability, the method involves propagating a large number of components, which may be costly in practice. Empirical analysis of runtime would be valuable.

Minor weaknesses:
The legend in Figure 1 lacks clarity
The LLMs work at the subword token level, which can lead to fragmented attributions for multi-token words. Do the authors aggregate subword tokens and how does this affect the results?

---

> ### Author Rebuttal · Authors · 2025-07-30
>
> Thanks for the valuable comments from the reviewer, we want to deliver further explanations accordingly.
>
> ### On the effectiveness of fixing nonlinearities
>
> - Freezing attention scores and MLP activations is **central to DePass's design**, enabling **lossless additive decomposition**. This allows us to precisely decompose and track contributions of hidden state components, ensuring their sum exactly reconstructs the original hidden state. This approach **eliminates second-order effects** during forward propagation. Without this freezing, dynamic internal computations would lead to non-linear, interactive attributions, undermining DePass's core advantages of additive decomposition and exact reconstruction.
>
> -  DePass's limitations don't stem from freezing these components, but from **not explaining attention scores** and **not modeling QK-OV circuit interactions**. As noted in Appendix F, freezing attention primarily captures information flow via the output-value (OV) circuit, **excluding query-key (QK) circuit interactions**. Decomposing QK circuits is a complex open problem due to their entangled nature and the quadratic complexity it would introduce. Similarly, dynamic MLP activations would compromise DePass's additive properties and exact reconstruction guarantees.
>
> - We'll clarify this rationale in the main paper, emphasizing it as a necessary trade-off for achieving lossless additive decomposition and faithfulness. We'll also expand on this limitation, discussing how future work might address QK circuit decomposition and dynamic MLP interactions while retaining DePass's core benefits.
>
> ### Empirical comparison of softmax (Eq. 14) and alternative functions
>
> - To evaluate the design choice of using softmax for MLP attribution, we compare it against two alternatives: a Linear-norm normalization (subtracting the minimum and dividing by the sum) and a Linear-weighted decomposition (linearly decomposing the contribution to the original activation value for each element). This comparison assesses whether the softmax rule is empirically justified.
>
> - Using the `known_1000` dataset with the `llama-2-7b-chat-hf` model, we report the results for both patch-top and recover-top token removal strategies:
>
> | Patch-Top   | 0.1       | 0.2       | 0.3       | 0.4       | 0.5       | 0.6       | 0.7       | 0.8       | 0.9       | 1.0       |
> | ----------- | --------- | --------- | --------- | --------- | --------- | --------- | --------- | --------- | --------- | --------- |
> | **Softmax** | **80.79** | **87.08** | **94.02** | **96.64** | **98.20** | **98.80** | **98.79** | **99.11** | **99.20** | **99.09** |
> | **Linear-norm**  | 64.80     | 73.64     | 85.73     | 93.17     | 96.55     | 97.46     | 98.17     | 98.77     | 99.03     | 99.09     |
> | **Linear-weighted decomposition**  | 59.37     | 66.84     | 77.53     | 86.20     | 92.42     | 94.98     | 96.62     | 98.38     | 98.60     | 99.09     |
>
> | Recover-Top | 0.1       | 0.2       | 0.3       | 0.4       | 0.5       | 0.6       | 0.7       | 0.8       | 0.9       | 1.0      |
> | ----------- | --------- | --------- | --------- | --------- | --------- | --------- | --------- | --------- | --------- | -------- |
> | **Softmax** | **97.81** | **96.87** | **93.92** | **89.89** | **81.57** | **75.50** | **68.94** | **56.84** | **43.50** | **0.00** |
> | **Linear-norm**  | 98.26     | 97.11     | 94.77     | 92.25     | 87.67     | 83.46     | 78.18     | 68.84     | 54.90     | 0.00     |
> | **Linear-weighted decomposition**  | 98.31     | 97.73     | 96.60     | 95.30     | 92.30     | 90.42     | 86.65     | 80.73     | 71.63     | 0.00     |
>
> - Results show that the softmax-based attribution consistently outperforms the alternatives across both token removal strategies, supporting its empirical effectiveness despite its heuristic origin.
>
> ### Clarification for selection of baselines
>
> - Our decision not to directly compare DePass against activation patching and general ablation-based attribution methods as performance baselines stems from two key reasons:
>
> - **Ablation as a Fidelity Metric (Ground Truth):**
>     - In mechanistic interpretability, ablation is often regarded as the **gold standard for evaluating attribution faithfulness** [1][2][3]. We therefore use ablation as our **ground truth for validation**, not as a competitive baseline.
>     - Our paper extensively demonstrates that DePass faithfully identifies critical components whose disruption or preservation, based on our scores, indeed impacts model behavior. Comparing DePass directly with ablation as if they perform identical tasks would be conceptually misaligned; **DePass provides attributions, and ablation validates them.**
>
> - **Computational Efficiency and Applicability, with Consideration for Activation Patching:**
>     - DePass is a direct and unified feature attribution framework based on a **single decomposed forward pass**, offering **significant computational advantages** over exhaustive, "computationally expensive" ablation methods which provide limited insight into intermediate information flow.
>     - Regarding activation patching methods, they typically require constructing specific contrastive datasets for activation replacement, which introduces additional complexity and computational overhead. Since ablation can be viewed as a specialized form of activation patching—specifically, patching with zero values—we did not assess it separately as a baseline.
>
> ### Illustration for multi-token word's attribution
> - We provide two examples of `llama-2-7b-chat-hf` below to illustrate how subword tokenization affects DePass scores. For word-level scores, we treat corresponding subword tokens as a **single component from the start of the DePass process**, not a simple summation of individual scores.
>
> **Example 1:**
>
> | Token     | `<s>` | `Catal` | `onia` | `belongs` | `to` | `the` | `continent` | `of` |
> | --------- | ----- | ------- | ------ | --------- | ---- | ----- | ----------- | ---- |
> | **Score** | 0.28  | 6.16    | 1.27   | 1.75      | 1.16 | 1.15  | 13.06       | 1.36 |
>
> | Word      | `<s>` | `Catalonia` | `belongs` | `to` | `the` | `continent` | `of` |
> | --------- | ----- | ----------- | --------- | ---- | ----- | ----------- | ---- |
> | **Score** | 0.28  | 6.38        | 1.92      | 1.37 | 1.38  | 13.24       | 1.59 |
>
> **Example 2:**
>
> | Token     | `<s>` | `The` | `mother` | `tongue` | `of` | `Daniel` | `le` | `Dar` | `rie` | `ux` | `is` |
> | --------- | ----- | ----- | -------- | -------- | ---- | -------- | ---- | ----- | ----- | ---- | ---- |
> | **Score** | 0.30  | 0.87  | 2.09     | 9.69     | 1.23 | -0.01    | 1.31 | 2.00  | 2.69  | 1.36 | 0.73 |
>
> | Word      | `<s>` | `The` | `mother` | `tongue` | `of` | `Danielle` | `Darrieux` | `is` |
> | --------- | ----- | ----- | -------- | -------- | ---- | ---------- | ---------- | ---- |
> | **Score** | 0.30  | 1.12  | 2.29     | 9.85     | 1.52 | 1.85       | 4.12       | 1.18 |
>
> - Both token-level and word-level attributions with DePass are effective. While token-level scores might disperse across subword tokens, they remain highly informative for distinguishing important tokens. Higher scores on specific subwords can even indicate triggers for the model's memory of the full word. DePass naturally supports aggregating subword tokens for desired word-level attribution.
>
> - To ensure a fair comparison with other token-level baselines, in the experiments corresponding to Figure 1, we perform ranking and ablation at the subword token level. We will clarify this in the revised text and further elaborate on the impact of subword tokenization on DePass.
>
> [1] Ameisen, et al., "Circuit Tracing: Revealing Computational Graphs in Language Models", Transformer Circuits, 2025.
>
> [2] Sparse Feature Circuits: Discovering and Editing Interpretable Causal Graphs in Language Models
>
> [3] Finding Neurons in a Haystack: Case Studies with Sparse Probing

---

> > ### Comment · Area_Chair_tjzg · 2025-08-05
> > **Please engage in discussion with authors**
> >
> > Dear Reviewer w5Wf,
> >
> > Thank you for your time and effort in reviewing this manuscript. The authors have prepared their rebuttal; can you please respond to it and engage in discussion with them? The discussion period will close on August 6th.
> >
> > Best,
> > AC

---

> > ### Comment · Reviewer_w5Wf · 2025-08-05
> >
> > I appreciate the time that authors took for all the extra experiments and clarifications. Adding such discussions and results in the paper will add for clarity and strengthen the paper. I still believe the paper is above threshold for acceptance and I maintain my rating.

---

### Official Review · Reviewer_D8ba · 2025-07-03

**Clarity:** 2
**Significance:** 3
**Originality:** 3
**Rating:** 4
**Confidence:** 3

**Summary:**

Forward pass an example through a transformer, then hold any nonlinearities (attention scores and MLP activations) fixed. With these static, the state of the residual stream at any layer can be broken down into components that sum to the hidden state, such that these components can be propagated forward through the network and still yield components at a later layer that correctly sum to the later hidden state. In essence, this allows attribution (e.g. breaking down the input by token, then attributing to each token the degree of influence it had on the output). Empirically, this seems to identify reasonable attributions on various tasks.

**Questions:**

1. In lines 63-66, what is "position $i$"? Is there any reference for the MLP layers' purpose being "to enhance per-token representation capacity"?
2. Why perform the softmax in Eq. 14?
3. What are the settings under which Fig. 1 is collected?
4. Why is DePass-abs introduced in Sec. 4.2 with no commentary? Intuitively, DePass-abs might make better sense as the "default" version of the technique.
5. In lines 245-249, why is layer 15 chosen for the decomposition projection?

**Ethical Concerns:**

["NO or VERY MINOR ethics concerns only"]

**Final Justification:**

I raised my score from 3 to 4.

As other reviewers pointed out, there are a number of comparisons to other interpretability techniques that are omitted from the work, which is a very significant limitation. However, the experiments are sufficient to demonstrate that the method is picking up on *some* important signals, and the authors' provided computational-efficiency suggested to me that this work at least occupies a place on the Pareto frontier of the expense-information density tradeoff of interpretability techniques. This is the type of work that I would be eager to see follow-ups of, so it seems suitable for the conference despite its limitations.

**Limitations:**

yes

**Quality:**

3

**Strengths And Weaknesses:**

## Strengths
1. The method appears to work empirically. Compared to other attribution methods, DePass appears able to more effectively identify key input tokens that change the outputs.
2. The authors highlight useful practical applications, e.g. measuring the truthfulness of a model.

## Weaknesses
1. Fixing nonlinearities intuitively might take away much of the representational power of a network (e.g. if a particular token's contribution is not on its own, but through allowing a different token to attend to it, like in the IOI task). Should we expect this, theoretically, to limit the method's effectiveness?
2. Certain parts of the explanation were weak (e.g. definition of neuron was hard to follow).
3. The paper would benefit from having more examples. Table 2 was helpful, but insufficient.
4. The key metric seems to be the change from masking, but additional detail would help here -- for instance, the distribution of attribution between different inputs.
5. Certain aspects of notation and labeling are imprecise (e.g. the various O variables, percents vs. fractions)
6. The method's value-add could use more clarification.
  - For instance, compared to ablation-based attribution, how much does DePass help computationally? As DePass still seems to require a forward pass for each "hypothesis" tested, it may not help much -- though without needing to compute attention weights, perhaps the computational complexity gain is immense.
  - Another example, why *can't* SDL capture evolution of a representation through a forward pass, via multilayer methods like the crosscoder.
  - (More generally, the claim that DePass tracks evolution through the forward pass could use more examination -- presumably all the intermediates along the residual stream are illegible sequences of numbers until the final decoding stage)

---

> ### Author Rebuttal · Authors · 2025-07-30
>
> Thanks for the elaborate comments, we will address them according to their proposed order.
>
> ### On the effectiveness of fixing nonlinearities
>
> - DePass is not concerned with the model's representational ability. Instead, its goal is to perform attribution analysis on a given forward pass. Freezing the nonlinearities allows us to precisely reconstruct the original forward pass. DePass ensures the target model's representational power is unaffected by using attention scores and MLP activations directly from its forward pass. This aligns with DePass's core principle: **efficient attribution with full fidelity to the original computation**. Our method specifically attributes **direct paths of information flow to the final logits**.
>
> - While some tasks, like Indirect Object Identification (IOI), involve indirect influences via **QK circuits**, capturing these complex interactions (a challenging open problem in mechanistic interpretability [1] ) is beyond our current scope. We believe attributing with QK circuits is a crucial future direction.
>
> ### Clarifying the definition of "neuron"
>
> - Following prior work [2], each MLP neuron is defined as a subvalue $v_k^\ell$, as introduced in Section 2 (Feedforward Network). For token $i$, the coefficient of neuron $k$ in layer $\ell$ is computed by corresponding subkey:
>
>     $$ m_{i,k}^\ell = \sigma(f_k^\ell \cdot {X}_{{attn},i}^\ell) $$
>     and output of the neuron is:
>
>     $$ Neuron_{i,k}^\ell = m_{i,k}^\ell \cdot v_k^\ell $$
>     The full MLP output is the sum over neurons' outputs:
>
>     $$ MLP_i^\ell = \sum_k Neuron_{i,k}^\ell$$
>     We will explicitly include a clarification in the future revisions.
>
> ### Additional examples and detail of our results
>
> - We agree that more examples are crucial for demonstrating DePass's capabilities and will expand relative section in the revision.
> - We provide a concise example illustrating the attribution distribution across inputs, including DePass (our method based on decomposed forward pass attribution), Normalized DePass (DePass scores normalized to sum to 1), All (Mean Attention), Last (Last-layer Attention), Rollout (Attention Rollout), Integrated Gradients (Integrated Gradients), Signed (Input × Gradient), and Norm (Gradient SHAP).
>
> **Prompt: "The capital of France is" Target: "Paris"**
> |Token|DePass|Normalized DePass|All|Last|Rollout|Integrated Gradients|Signed|Norm|
> |:-|:-|:-|:-|:-|:-|:-|:-|:-|
> |`<s>`|-0.30|-0.02|0.30|0.28|0.35|0.00|0.17|0.40|
> |`The`|1.20|0.07|0.14|0.14|0.13|0.01|0.17|0.14|
> |`capital`|8.94|0.52|0.14|0.14|0.13|0.84|0.16|0.17|
> |`of`|1.02|0.06|0.14|0.14|0.13|0.14|0.17|0.14|
> |`France`|5.06|0.30|0.14|0.15|0.13|0.00|0.16|0.07|
> |`is`|1.20|0.07|0.14|0.15|0.13|0.01|0.17|0.09|
>
> **Prompt: "The capital of France is" Target: "known"**
> |Token|DePass|Normalized DePass|All|Last|Rollout|Integrated Gradients|Signed|Norm|
> |:-|:-|:-|:-|:-|:-|:-|:-|:-|
> |`<s>`|0.09|0.01|0.30|0.28|0.35|0.26|0.18|1.00|
> |`The`|1.91|0.15|0.14|0.14|0.13|0.00|0.08|0.00|
> |`capital`|6.00|0.48|0.14|0.14|0.13|0.65|0.20|0.00|
> |`of`|1.41|0.11|0.14|0.14|0.13|0.00|0.16|0.00|
> |`France`|1.80|0.14|0.14|0.15|0.13|0.00|0.22|0.00|
> |`is`|1.41|0.11|0.14|0.15|0.13|0.09|0.17|0.00|
>
> **Prompt: "grape: purple, banana: yellow, apple:" Target: "red"**
> |Token|DePass|Normalized DePass|All|Last|Rollout|Integrated Gradients|Signed|Norm|
> |:-|:-|:-|:-|:-|:-|:-|:-|:-|
> |`<s>`|0.32|0.02|0.14|0.12|0.17|0.02|0.07|0.88|
> |`gra`|1.52|0.11|0.07|0.07|0.06|0.00|0.06|0.00|
> |`pe`|0.08|0.01|0.07|0.07|0.06|0.00|0.10|0.00|
> |`:`|0.20|0.01|0.07|0.07|0.06|0.00|0.06|0.11|
> |`pur`|5.25|0.36|0.07|0.07|0.06|0.00|0.07|0.00|
> |`ple`|0.14|0.01|0.07|0.07|0.06|0.00|0.08|0.00|
> |`,`|0.10|0.01|0.07|0.07|0.06|0.00|0.06|0.00|
> |`ban`|0.48|0.03|0.07|0.07|0.06|0.00|0.08|0.00|
> |`ana`|0.12|0.01|0.07|0.07|0.06|0.00|0.05|0.00|
> |`:`|0.15|0.01|0.07|0.07|0.06|0.98|0.09|0.00|
> |`yellow`|1.53|0.11|0.07|0.07|0.06|0.00|0.05|0.00|
> |`,`|0.19|0.01|0.07|0.07|0.06|0.00|0.08|0.00|
> |`apple`|4.16|0.29|0.07|0.07|0.06|0.00|0.08|0.00|
> |`:`|0.17|0.01|0.07|0.07|0.06|0.00|0.06|0.00|
>
> - DePass reports both raw and normalized attribution scores, with raw values summing exactly to the output logit. This allows direct interpretation of each score’s impact on the prediction. For example, when comparing targets like *“Paris”* and *“known”* for the same input, their total attribution scores differ.
> - DePass attributes the output to semantically meaningful parts of the input, while baseline and gradient-based methods often fail to emphasize relevant tokens or align with clear semantic patterns.
>
> ### Clarification for certain notation and labeling
> - We appreciate your feedback regarding notation precision, especially concerning the "O" variables and the clarity of "percents vs. fractions" in our figures.
>
> ##### Clarification of "O" Notation
> - Regarding the "O" symbols, their usage is consistent but we understand the need for clearer definitions. We'll refine these in the revision:
>     - **$O^{l,j}$** (Eq. 2): output of the $j$-th attention head in layer $l$, representing overall attention without token-wise decomposition.
>     - **$o_{i,m}^{(l,j)}$** (Eqs. 12,13): output of the $j$-th attention head at layer $l$ for the $m$-th decomposed component of token $i$ in DePass, showing propagation through attention.
>     - **$o_{i}^{(\ell,h)}$** (Eq. 20): output of the $h$-th attention head at layer $l$ for token $i$ in attention decomposition, used to initialize decomposed states for attribution.
>
> ##### Units in Figures
>
> - We acknowledge that the axes in Figures 1 and 3 require more explicit labeling. We will update the labels and captions to clearly indicate units:
>
>     - **Figure 1**: x-axis is masking *proportion* (0 to 1); y-axis is average relative probability change in percentage (0 to 100).
>     - **Figure 3**: both x-axis (mask percentage) and y-axis (accuracy) range from 0 to 100.
>
> ##### "Position" in lines 63-66
> - We'll be more precise with new terminology to avoid confusion. "Position" specifically refers to each token's location within the sequence.
>
> ### Clarification for DePass's value-add
>
> ##### Efficiency and Parallel Computation
>
> - DePass's **parallel computation** greatly enhances efficiency. It enables the simultaneous attribution of many hypotheses of the same type, such as all MLP neurons in a layer, in just **one forward decomposition pass**. This contrasts sharply with ablation, which demands individual masking and re-evaluation for each neuron, leading to considerably higher computational overhead.
> - The table below demonstrates DePass's runtime advantage for neuron-level attributions over an intermediate MLP layer in different LLaMA models. The current implementations already show strong performance, with considerable potential for further acceleration.
>
> |Model|Intermediate Size (number of neurons)|Method|Time (s)|
> |:-|:-|:-|:-|
> |LLaMA-2-7B-Chat|11008|Ablation|321.04|
> |||DePass|**7.22**|
> |LLaMA-3.2-3B-Instruct|8192|Ablation|234.77|
> |||DePass|**2.91**|
> |LLaMA-3.2-1B-Instruct|8192|Ablation|134.76|
> |||DePass|**2.23**|
>
> - Moreover, for studying the role of subspaces, ablation is inherently limited. Naively ablating a subspace during execution can be misleading, as subsequent layers may compensate for the missing information [3]. In contrast, DePass allows us to directly trace and quantify the contribution of a specific subspace without perturbing the network's forward dynamics.
>
> ##### Further discussion about SDL-based methods
>
> - Given space constraints, we direct you to our response to reviewer uE26 for further details.
>
> ### References for MLP layers' purpose
> - Key references on MLPs enhancing per-token representation capacity include [4-6]
>
> ### Empirical comparison of softmax (Eq. 14) and alternative functions
> - Given space constraints, we direct you to our response to reviewer w5Wf for further details.
>
> ### Settings of results in Fig.1
> - For each data point in the `Known_1000` and `IOI` datasets using the `Llama-2-13b-chat-hf` model, we:
>     * Obtain attribution scores for the correct answer via various methods.
>     * Remove tokens from the prompt based on these scores, following either a "patch top" (highest attribution) or "recover top" (lowest attribution) strategy.
>     * Reassemble the remaining tokens into a new prompt and rerun the model.
>     * Measure the relative probability change of the correct answer.
> - The figures show the average relative probability change across all data points per dataset at each masking level.
>
> ### Clarification for introducing DePass-abs
> - In lines 201-211, we introduced DePass-Abs as the absolute values of DePass scores, capturing both supportive and suppressive contributions.
> - We chose not to use DePass-Abs as the default to preserve the **direction** of contributions. Raw DePass scores reflect signed contributions, distinguishing positive support from negative suppression—critical for interpreting token-level and subspace-level effects, such as identifying inputs driving truthful versus untruthful outputs.
> DePass-Abs, which ignores direction, is used at the component level where negative contributions can still be functionally important.
>
> ### Clarification for layer selection in Section 4.3
> - We chose Layer 15 for its strong probing performance and effective separation of language-specific signals, as demonstrated empirically. While other intermediate layers (13-25) exhibit similar effects, Layer 15 serves as a representative example.  We'll include results from additional layers in the revised version.
>
> [1] Circuit Tracing: Revealing Computational Graphs in Language Models, Transformer Circuits.
>
> [2] Neuron-Level Knowledge Attribution in Large Language Models
>
> [3] Is this the subspace you are looking for? An interpretability illusion for subspace activation patching
>
> [4] Dissecting Recall of Factual Associations in Auto-Regressive Language Models
>
> [5] Transformer Feed-Forward Layers Are Key-Value Memories
>
> [6] Locating and Editing Factual Associations in GPT

---

> > ### Comment · Reviewer_D8ba · 2025-08-02
> >
> > Thank you very much for the detailed response.
> >
> > ## On the effectiveness of fixing nonlinearities
> > I have follow-up questions here:
> > 1. What kinds of problems will be very hard for DePass to solve due to relevant action happening in the nonlinearities? Either QK or perhaps in linear activations.
> > 2. Can we anticipate any ways in which DePass might be useful as a foundation for decompositions that account for nonlinearities?
> >
> > ## Clarifying the definition of "neuron"
> > ## Clarification for certain notation and labeling
> > Thank you for making these clarifications.
> >
> > ## Additional examples and detail of our results
> > I appreciate these tables a lot; they are very information-dense.
> >
> > ## Clarification for introducing DePass-abs
> > I am curious about a version of the above tables that includes a strong negative DePass score. However, my concerns here are assuaged; it seems logical to minimize the complexity of the method and leave the absolute value out of the “default” version (so, I do not ask that you make such a table for the review process). I do think at least one sentence about why DePass-abs is useful for this setup but not the other ones would be useful though.
> >
> > ## Clarification for DePass's value-add
> > Thank you for the table; this seems to pretty conclusively show the advantage of DePass over ablation-based methods in terms of computational efficiency. However, I think the ablation result is more useful in certain ways for determining actual network performance – the network’s compensation for the ablation of particular features is not a behavior we want to ignore! Still, I agree that DePass still has value here, in that it theoretically allows one to follow how a particular representation is modified through a forward pass.
> >
> > I think one question remains still: does it matter that we can follow how this representation is modified through a forward pass? Is it not still an uninterpretable jumble of floats until the final layer?
> >
> > ## Further discussion about SDL-based methods
> > I think the claim that SAE-based methods haven’t “been systematically evaluated across datasets” misses the point – if we want to demonstrate DePass’s value, we should try to get a direct comparison and show that DePass is similarly useful on some downstream task, while being computationally more efficient/more scalable/better tested, etc. We mainly care about interpretability tools by the extent to which they actually help us interpret networks.
> >
> > I’m not necessarily calling for such an evaluation here – I realize that the costs of training SAEs are very high. Still, the “feature evolution” tracking in an SAE is arguably the more interesting kind – tracking what abstractions the model is using and building with to get to an answer, rather than just tracking the particular embedding vector that’s passed through layer-by-layer. Even if it’s lossy, it certainly seems useful.
> >
> > So, I think my question remains here. Is there something practically useful that DePass could potentially do that SAEs couldn’t, in terms of tracking evolution of a representation through a forward pass? The latter half of the question based on the claim in lines 287-288 – I certainly agree that DePass is more scalable than SAEs.
> >
> > ## References for MLP layers' purpose
> > Thank you for these references.
> >
> > ## Empirical comparison of softmax (Eq. 14) and alternative functions
> > Thank you for the clarification. I did not see where it was indicated that this softmax was primarily empirical/heuristic. It would be worth noting this and including the results table from your response to Reviewer w5Wf in the Appendix, perhaps.
> >
> > ## Settings of results in Fig.1
> > Thank you for the clarification.
> >
> > ## Clarification for layer selection in Section 4.3
> > Thank you for the clarification. I think this choice would be well served with a figure or table in the Supplementary Materials justifying it.

---

> > > ### Author Response · Authors · 2025-08-04
> > >
> > > We appreciate the timely response and the highly professional and inspiring feedback. Below we provide our further discussion.
> > >
> > > ### On the effectiveness of fixing nonlinearities
> > >
> > > - We do not identify any class of problems for which DePass would be fundamentally inapplicable. Any influence on the model’s output must ultimately propagate through the OV pathway, and DePass captures these direct contributions exactly. The main limitation is that in settings where the goal is to identify indirect contributors (e.g., multi-step reasoning or IOI-like tasks), the direct sources surfaced by DePass may differ from the components of interest.
> > >
> > > - DePass can indeed serve as a solid foundation for richer decomposition methods. The DePass formulation already accommodates attention-score attribution: for two decomposed states, projecting through QK and multiplying recovers the exact original attention weights, enabling attribution of which component interactions drove score formation. This can be combined with OV-path attribution for a more complete view. As such, DePass provides a natural basis for extensions that integrate nonlinearity effects — for example, by jointly analyzing QK- and OV-side contributions. For MLP decomposition, we have not identified any fundamental limitations in the current approximation, though further refinement could potentially yield even better results.
> > >
> > > ### Clarification for introducing DePass-abs
> > >
> > > - In token-level attribution, strong negative scores are rare; tokens unrelated to the output typically score near zero. In component-level decomposition, DePass often detects components with substantial negative contributions. We introduce DePass-abs to capture such components. For ablation, where ranking is required, DePass-abs sort by the absolute value to ensure both positive and negative contributors are considered. We will clarify in the paper.
> > >
> > > ### Clarification for DePass's value-add
> > >
> > > - We agree that ablation-based methods are valuable for certain research purposes. We appreciate the inspiring feedback. DePass offers a complementary interpretability perspective, as model behavior is complex and benefits from multiple viewpoints.
> > > - While DePass does not directly attach semantic meaning to these changes, it integrates naturally with interpretability tools such as probing classifiers or SAEs, which align hidden states with human-interpretable concepts. This allows one to track not just the raw vector evolution but also the evolution of semantically meaningful features, combining the precision of DePass with the interpretability of feature-alignment methods.
> > >
> > > ### Further discussion about SDL-based methods
> > >
> > > - We agree with your rigorous comparative perspective. Although a full evaluation would be complex, we will consider conducting such experiments in future work.
> > > - We appreciate your acknowledgment of DePass’s scalability. SAE is an important and widely used interpretability method. Both SAE and DePass decompose the residual stream; therefore, for a given model and its corresponding SAE, they can, in principle, track the same content, although SAE would require additional computation over the indirect influence matrix [1] between features. For a model in training, DePass can observe changes in feature evolution throughout the training process. We will revise the claim in lines 287–288 accordingly. Moreover, DePass can be combined with SAE to provide semantically interpretable feature-evolution tracking, often in a way that is more direct and convenient than graph-based approaches.
> > >
> > > ### Other Clarifications
> > >
> > > - Empirical comparison of softmax and alternative functions: We will clarify our motivation for using softmax in the main text and include the empirical comparison in the appendix.
> > > - Layer selection in Section 4.3: We will provide a more detailed explanation of the layer selection process and add the corresponding data in the appendix.
> > >
> > > [1] Circuit Tracing: Revealing Computational Graphs in Language Models, Transformer Circuits.

---

> ### Comment · Reviewer_D8ba · 2025-08-04
>
> > The main limitation is that in settings where the goal is to identify indirect contributors (e.g., multi-step reasoning or IOI-like tasks), the direct sources surfaced by DePass may differ from the components of interest.
>
> This does seem like the concern in such settings. By construction, DePass will always produce internally consistent and complete decompositions. The question is whether DePass's decompositions will be useful on closer-to-practical tasks, and that hasn't been addressed.
>
> > While DePass does not directly attach semantic meaning to these changes, it integrates naturally with interpretability tools such as probing classifiers or SAEs, which align hidden states with human-interpretable concepts.
> > Moreover, DePass can be combined with SAE to provide semantically interpretable feature-evolution tracking, often in a way that is more direct and convenient than graph-based approaches.
>
> Ah, this seems quite useful to explore! If indeed DePass's subspace decomposition pulls out linear probe-identified directions or isolates SAE features, that would serve as useful validation that these independent methods are pointing at some meaningful underlying reality in models' representation-spaces. I would be very excited to see this work done, but it seems unrealistic to ask for this in the review process.
>
> ---
>
> I intend to raise my score from a 3 to a 4. I think the work is meaningful and worth accepting, particularly with the improved language and example coverage for clarity. There are still important axes of evaluation which, if done, could stand to validate the work as *practically applicable*, and in their absence, I think a 4 is appropriate.

---

> > ### Author Response · Authors · 2025-08-05
> >
> > Thank you for your engagement and valuable feedback.
> >
> > We will revise the paper and include additional examples, making the explanations clearer. We are also excited to explore the combination of DePass with SAE and apply it to more practical scenarios in future work.

---

### Official Review · Reviewer_uE26 · 2025-07-03

**Clarity:** 3
**Significance:** 3
**Originality:** 3
**Rating:** 3
**Confidence:** 2

**Summary:**

The paper presents a novel method for interpreting how tokens, modules, or subspace components influence the output, which can be used for output attribution and understanding processes inside model. The method decomposes the hidden state into m components, while the original hidden state can be recovered by summing up the components.

**Questions:**

- Is it possible to apply these method for the model steering?
- Can we combine DePass with SAE by decomposing input into desired feature subspace and other components? Or apply it token-wise, like in Eq. 19, for vectors from the SAE decoder?

Also see weaknesses.

**Ethical Concerns:**

["NO or VERY MINOR ethics concerns only"]

**Final Justification:**

After carefully reading the authors’ response, I have decided to maintain my score and remain slightly inclined toward rejection. I would like to acknowledge the experiments conducted during the rebuttal and discussion phases, which demonstrate the potential of integrating DePass into MI research. However, these experiments are largely qualitative in nature and lack quantitative validation. Regarding the paper as it stood before the rebuttal, DePass appeared somewhat isolated, with limited connection to existing methods. I believe this work could significantly benefit from further refinements aligned with the promising direction outlined during the discussion phase.

**Limitations:**

See W3. From my perspective, the main limitation is the computational resources required to process m components through the model, which results in approximately m times the original FLOPs count.

**Paper Formatting Concerns:**

There are no major formatting issues in this paper.

**Quality:**

2

**Strengths And Weaknesses:**

Strengths

- S1: The method can be applied to pretrained language models without requiring any additional training.
- S2: The method achieves superior performance across a variety of tasks.
- S3: The paper is well-written.

Weaknesses

- W1: The method incurs significant overhead during inference, particularly when applied to token-level attribution tasks.
- W2: The authors argue that the features identified by SAE do not specify which token activates a particular feature. However, there are several works that explore computational graphs [1] and later studies [2], based on transcoders or crosscoders. While these methods require computationally intensive training, the authors could provide justification for these approaches or include a comparison in the context of token-level attribution. This is particularly relevant given the availability of open-source transcoders for large models such as LLaMA-3.1-8B [3].
- W3: No limitations of the method discussed in paper.

Other comments:

The authors state that SAE-based methods "do not capture how specific features evolve throughout the forward pass." However, there are methods such as [4], [5], and [6] that discover the evolution of SAE features during the forward pass.

[1] [Transcoders Find Interpretable LLM Feature Circuits](https://proceedings.neurips.cc/paper_files/paper/2024/file/2b8f4db0464cc5b6e9d5e6bea4b9f308-Paper-Conference.pdf)

[2] Ameisen, et al., "Circuit Tracing: Revealing Computational Graphs in Language Models", Transformer Circuits, 2025.

[3] [Llama Scope: Extracting Millions of Features from Llama-3.1-8B with  Sparse Autoencoders](https://arxiv.org/pdf/2410.20526)

[4] [Mechanistic Permutability: Match Features Across Layers](https://openreview.net/forum?id=MDvecs7EvO)

[5] [Analyze Feature Flow to Enhance Interpretation and Steering in Language Models](https://arxiv.org/pdf/2502.03032)

[6] [Evolution of SAE Features Across Layers in LLMs](https://arxiv.org/pdf/2410.08869)

---

> ### Author Rebuttal · Authors · 2025-07-30
>
> Thanks for the detailed comments from the reviewer, we would like to make several clarifications below.
>
> ### DePass's Efficiency & Fidelity Trade-off
>
> - We acknowledge that DePass introduces additional computational cost compared to a single standard forward pass. However, this reflects a deliberate trade-off between efficiency and **attribution accuracy**. Prior methods often prioritize low computational cost but sacrifice attribution **fidelity**, especially in deep, nonlinear models like LLMs. In contrast, DePass maintains the integrity of the original forward process, producing higher-fidelity, more reliable attributions within an acceptable runtime budget.
>
> - The table below demonstrates DePass's runtime advantage for neuron-level attributions over an intermediate MLP layer in different LLaMA models compared to ablation method. The current implementations already show strong performance, with considerable potential for further acceleration.
>
> |Model|Intermediate Size (number of neurons)|Method|Time (s)|
> |:-|:-|:-|:-|
> |LLaMA-2-7B-Chat|11008|Ablation|321.04|
> |||DePass|**7.22**|
> |LLaMA-3.2-3B-Instruct|8192|Ablation|234.77|
> |||DePass|**2.91**|
> |LLaMA-3.2-1B-Instruct|8192|Ablation|134.76|
> |||DePass|**2.23**|
>
> ### Further discussion about SAE-based methods
> We thank the reviewer for pointing out the lack of clarity in our discussion of SAE. We will revise the paper to improve clarity and **include additional citations [2][4][6][7]**.
>
> ##### On which token activates particular feature
> - We appreciate the reviewer for highlighting prior work that leverages computational graphs and transcoders to trace which token activates a particular feature. We acknowledge the validity of these approaches and will revise the wording in our paper to reflect this more accurately.
>
> - While these methods analyze token-feature activation, they often involve **complex and hard-to-scale graph computation and pruning**. To our knowledge, their token-level attribution **fidelity hasn't been systematically evaluated across datasets**.
>
> - In contrast, DePass offers a **simple, scalable, and direct attribution framework**. Instead of proxy measures like token-feature correlations, DePass preserves and propagates attribution information throughout the **actual forward process**, enabling accurate token-level attribution without extra training or complex graph tracing. This practicality and fidelity make DePass more suitable for widespread use in large-scale model analysis.
>
> ##### On Feature Evolution Tracking
> - We appreciate you raising this important point. We'll leverage the terminology from [1] for further discussion. We acknowledge that existing work on transcoders and crosscoders can track abstract features across layers within their trained **'Isomorphic Models'**. However, when DePass refers to "feature evolution," it denotes **precise tracking within the 'Literal Model.'** We’ll refine our phrasing and provide a more in-depth discussion in the revised version, along with additional relevant references.
> - DePass's "feature evolution" or "tracking" precisely traces the information flow of a specific vector or subspace throughout the **literal model's computation graph**. It can accurately isolate the subspace that evolved from a target feature at any point in the residual stream.
> - Transcoder-based feature tracking, in contrast, relies on importance scores derived from correlations between sparsely trained encoders and decoders at different layers [2][3][4][5]. While these methods can identify "circuits" through careful filtering and pruning, reconstructing a feature's influence at a given point based on activation relationships is inherently lossy. Moreover, this tracking is in the context of **Isomorphic Models**, not the literal model.
> - Crosscoder-based feature tracking integrates sparsely trained encoders and decoders from different layers to identify features that remain relatively stable across layers in the residual stream. While this simplifies tracking shared features across layers within **Isomorphic Models**, it remains lossy in the context of the literal model.
> - Furthermore, work [6] indicates that semantic similarity between features isn't accurately reflected by directional similarity. Methods that derive causal relationships between cross-layer features from activation correlations across large corpora cannot perform feature evolution tracking in either sense.
> - Nevertheless, we believe DePass can serve as an effective bridge connecting analyses in **Isomorphic Models** with the **literal model**, which we'll discuss in the following section.
>
> ##### Capacity of combining DePass with SAE
> - DePass can be applied at the subspace level by treating the SAE features of interest as DePass components, which enables direct and convenient analysis of the interactions among SAE features across different layers.
> - Alternatively, DePass can be used at the token level, similar to Eq. 19, to investigate which tokens’ semantics jointly activate a particular SAE feature.
> - We are interested in conducting such experiments. However, due to the substantial annotation effort required to reliably label SAE features in datasets, we currently lack access to sufficiently annotated feature representations. Moreover, mapping SAE features to human-interpretable semantic spaces remains challenging, making it difficult to obtain ground-truth data for rigorous experimental validation at this time.
>
> ##### Capacity of combining DePass with model steering
>
> - DePass can be naturally applied to model steering analysis and offers a novel perspective for evaluating the effectiveness of steering vectors. By treating the steering vector as a DePass component during the forward pass, we can accurately quantify its influence on the model’s final output. Furthermore, when combined with tools like SAE or probing methods that analyze hidden states, DePass enables a comprehensive examination of how steering vectors affect the model’s internal dynamics across layers.
> - We believe there is substantial potential for further integrating DePass with model steering techniques to deepen interpretability and control in large language models.
>
> ### On Method Limitations
>
> - We appreciate the reviewer's feedback regarding method limitations. We do discuss the limitations of our method in the appendix F section of the paper. We will clarify this in the main text in the revision to ensure readers can easily find and understand the method’s limitations.
>
>
> [1] Sparse Crosscoders for Cross-Layer Features and Model Diffing
>
> [2] Ameisen, et al., "Circuit Tracing: Revealing Computational Graphs in Language Models", Transformer Circuits, 2025.
>
> [3] Transcoders Find Interpretable LLM Feature Circuits
>
> [4] Mechanistic Permutability: Match Features Across Layers
>
> [5] Llama Scope: Extracting Millions of Features from Llama-3.1-8B with Sparse Autoencoders
>
> [6] Analyze Feature Flow to Enhance Interpretation and Steering in Language Models
>
> [7] Evolution of SAE Features Across Layers in LLMs

---

> > ### Comment · Reviewer_uE26 · 2025-08-03
> >
> > I thank the authors for the detailed response. I will go point by point and address the points raised in the rebuttal.
> >
> > **DePass's Efficiency & Fidelity Trade-off**
> >
> > Thank you for providing additional results on the model’s speed.
> >
> > **Scaling Attribution Graph**
> >
> > > While these methods analyze token-feature activation, they often involve complex and hard-to-scale graph computation and pruning.
> >
> > In practice, the weights for the attribution graph can often be pre-computed. Many demos -- such as "Circuit Tracer" in Neuronpedia -- appear to compute these graphs efficiently on CPU. This suggests that the graph computation may be more scalable in applied settings than it might initially seem.
> >
> > **Combining DePass with SAE**
> >
> > I agree with the distinction between Isomorphic and Literal models. However, Isomorphic models can still be valuable for understanding and steering the behavior of Literal models using SAE, particularly since steering on downstream tasks can be easily evaluated (See, for example, [1.1]). While SAE provides a lossy compression of the hidden state, it brings important practical advantages. In many cases, methods with some information loss but better downstream performance are more likely to be adopted in practice. Therefore, combining DePass with standard MI methods, including SAE, is important for this work.
> >
> > > However, due to the substantial annotation effort required to reliably label SAE features in datasets, we currently lack access to sufficiently annotated feature representations.
> >
> > There are open-source annotated SAEs available for the LLaMA-3.1-8B model used in your experiments (see Neuronpedia), with coverage across all layers. Additionally, SAE representations are transferable from base to instruct models without the need for retraining [1.2].
> >
> > I am not suggesting the authors incorporate this into the current rebuttal, but it is worth noting that this can be done with publicly available tools and data.
> >
> > I appreciate the authors' proposals for combining SAE and DePass. These ideas can indeed expand the potential impact of the method.
> >
> > ----
> >
> > Overall, I thank the authors for their explanations and clarifications. I appreciate the effort involved in responding thoroughly to the concerns raised.
> >
> > References:
> >
> > [1.1] Improving Steering Vectors by Targeting Sparse Autoencoder Features. Sviatoslav Chalnev, Matthew Siu, Arthur Conmy.
> >
> > [1.2] SAEs (usually) Transfer Between Base and Chat Models. Connor Kissane, Robert Krzyzanowski, Arthur Conmy, Neel Nanda.

---

> > > ### Author Response · Authors · 2025-08-04
> > >
> > > - Discussion of Attribution Graph: We thank the reviewer for the information and resources, and will revise our discussion of attribution graphs accordingly in the revised paper.
> > > - Combining DePass with SAE: We agree that SAE is irreplaceable for aligning model representations with human-interpretable concepts. Combining DePass with SAE is therefore an important approach to bridge literal models and isomorphic models. DePass can flexibly connect various elements of a literal model—tokens, components, and subspaces—with SAE features. Below we provide a token‑level example illustrating the potential of combining DePass with SAE.
> > >
> > > `prompt`: "In exploring the relationship between climate change and urban planning, the research identifies key strategies for sustainable development. It underscores the urgency of integrating environmental considerations into city planning."
> > >
> > > `Model`: LLaMA‑3.1‑8B and corresponding SAE
> > >
> > > `Feature`: Layer 31, Feature 5226 – “climate change and its associated impacts” (annotation from Neuronpedia)
> > >
> > > We analyze three tokens that activate the target feature. DePass precisely identifies that the activation of this SAE feature is driven by semantic information originating from `climate` and propagating to other relevant tokens.
> > >
> > > Notes:
> > >
> > > - *climate (6)* indicates `climate` is the 7th token in the tokenized sequence.
> > > - Activation value is the pre‑activation score (dot product of the hidden state and feature weight).
> > > - Tokens with contribution magnitude < 0.1 are omitted.
> > >
> > >
> > > | Token analyzed (index)    | Activation value | Source token (index) | Contribution |
> > > | ----------------- | ---------------- | -------------------- | ------------ |
> > > | **climate (6)**       | 35.2500          | climate (6)         | 26.8750      |
> > > |                   |                  | In (1)               | 3.0781       |
> > > |                   |                  | exploring (2)       | 1.8750       |
> > > |                   |                  | the (3)             | 1.4844       |
> > > |                   |                  | relationship (4)    | 1.4844       |
> > > |                   |                  | between (5)         | 1.4844       |
> > > | **change (7)**        | 15.6875          | climate (6)         | 16.5000      |
> > > |                   |                  | relationship (4)    | 0.8594       |
> > > |                   |                  | between (5)         | 0.3984       |
> > > |                   |                  | change (7)          | 0.5586       |
> > > |                   |                  | In (1)              | -1.9141      |
> > > | **environmental (27)** | 7.4688           | climate (6)         | 9.4375       |
> > > |                   |                  | exploring (2)       | 1.0234       |
> > > |                   |                  | environmental (27)  | -0.6562      |
> > > |                   |                  | underscores (22)    | -0.1670      |
> > > |                   |                  | urgency (24)        | -0.2432      |
> > > |                   |                  | integrating (26)    | -0.3516      |

---

> > > > ### Comment · Reviewer_uE26 · 2025-08-09
> > > >
> > > > Thank you for including results on SAEs; this significantly enhances the practical utility of the work and facilitates the integration of DePass into existing mechanistic interpretability research. One limitation of these results, however, is that they are primarily qualitative rather than quantitative, which raises the possibility—however unlikely—of cherry-picking (though I have reasonable confidence this is not the case, we should aim to eliminate even the perception of such a risk).
> > > >
> > > > As an example of a more quantitative approach, I suggest an evaluation using a prompt such as: “Is feature [feature explanation] related to tokens [tokens with scores above threshold] in the example [text]?” This could then be compared to results from activation patching. My concern is that incorporating such experiments would require a substantial shift in the paper’s narrative and framing—changes that cannot be properly assessed during the rebuttal and discussion phase, since reviewers do not have access to revised PDFs.
> > > >
> > > > That said, if other reviewers advocate for acceptance, I am open to adjusting my score accordingly.
> > > >
> > > > I appreciate the authors’ responsiveness and openness during the rebuttal and discussion period. For now, I will maintain my current score and reserve final judgment until the next phase.

---

> > > > > ### Author Response · Authors · 2025-08-09
> > > > >
> > > > > We appreciate the reviewer’s thoughtful suggestion for a more quantitative evaluation. We agree this would further strengthen the work and have already begun planning such experiments for the final version. Meanwhile, we provide additional demonstrations here to further support our claims.
> > > > >
> > > > > `prompt`: "Avocado toast, a simple yet trendy breakfast option, gains a delightful twist with a sprinkle of chili flakes and a drizzle of honey for that perfect balance of spicy and sweet."
> > > > >
> > > > > `Feature`: Layer 31, Feature 26874 – “the context of simplicity in various contexts”
> > > > >
> > > > > | Token analyzed (index) | Activation value | Source token (index) | Contribution |
> > > > > | ---------------------- | ---------------- | -------------------- | ------------ |
> > > > > | **yet (7)**            | 7.4062           | simple (6)          | 5.8438       |
> > > > > |                        |                  | toast (3)           | 1.9219       |
> > > > > |                        |                  | yet (7)             | 1.1953       |
> > > > > |                        |                  | , (4)                | -0.5312      |
> > > > > |                        |                  | a (5)               | -0.5898      |
> > > > > |                        |                  | \<begin\_of\_text\> (0) | -1.3750     |
> > > > > | **,(11)**              | 3.5312           | simple (6)          | 1.7031       |
> > > > > |                        |                  | toast (3)           | 0.8125       |
> > > > > |                        |                  | yet (7)             | 0.7344       |
> > > > > |                        |                  | \<begin\_of\_text\> (0) | -1.8906     |
> > > > >
> > > > >
> > > > > `prompt`: "In economics, supply and demand is a model for understanding how prices and quantities are determined in a market system. This concept is foundational in economic theory and affects various market structures."
> > > > >
> > > > > `Feature`: Layer 25, Feature 9618 – “references to economic topics and concepts”
> > > > >
> > > > > | Token analyzed (index) | Activation value | Source token (index)   | Contribution |
> > > > > | ---------------------- | ---------------- | ---------------------- | ------------ |
> > > > > | **economic (28)**      | 15.0625          | economics (2)          | 5.9375       |
> > > > > |                        |                  | economic (28)          | 4.9062       |
> > > > > |                        |                  | \<begin\_of\_text\> (0) | 0.9805       |
> > > > > |                        |                  | . (22)                 | 0.1348       |
> > > > > | **and (30)**           | 7.3438           | economics (2)          | 4.7500       |
> > > > > |                        |                  | economic (28)          | 1.9844       |
> > > > > |                        |                  | \<begin\_of\_text\> (0) | 1.1719       |
> > > > > |                        |                  | concept (24)           | -0.1025      |
> > > > > |                        |                  | In (1)                 | -0.6680      |
> > > > >
> > > > > While we agree that adding the suggested quantitative evaluation would further strengthen the work, the paper already provides strong evidence supporting our main claims. As a training-free method, DePass demonstrates broad versatility across multiple attribution granularities—including tokens, model components, and subspaces—validated by extensive experiments. Other reviewers have also recognized these contributions as sufficient. Experiment combining DePass with SAEs is only a subset of its subspace applications, and we have demonstrated its effectiveness on language-specific subspaces. Therefore, despite the absence of the proposed quantitative experiment in the current version, we believe the main claims remain well justified.

---

### Note · Authors · 2025-08-12

We are grateful for the reviewers' thoughtful, timely, and constructive feedback. We are pleased that all reviewers recognize DePass as a powerful, training-free interpretability method with broad versatility.

During the rebuttal, we provided detailed explanations to address concerns about freezing non-linear operations and the MLP decomposition approach. We also clarified the efficiency benefits of DePass over ablation and included additional examples to improve clarity.

Our rebuttal has successfully clarified key aspects of our work and positively influenced the reviewers' assessments. Specifically, Reviewer w5Wf has reaffirmed the belief that our paper is "above threshold for acceptance". Reviewer D8ba has indicated an intention to raise score to a 4, stating "the work is meaningful and worth accepting." Reviewer uE26 has expressed an openness to adjusting score based on the consensus of the other reviewers.

The feedback we received is valuable. We are committed to incorporating the refinements discussed to further enhance the clarity and strength of our paper, and we are optimistic that DePass will make a valuable contribution to the field of interpretability.

---

### Decision · Program_Chairs · 2025-09-17

**Decision:**

Accept (poster)

**Comment:**

**Summary:**

The paper proposes a new mechanistic interpretability method enabling feature attribution with a single forward pass.

**Strengths:**

- Reviewers agree the method seems innovative and could add value to the community.
- The results are strong / promising
- “elegant and clearly described contribution” (w5Wf)
- “Compared to other attribution methods, DePass appears able to more effectively identify key input tokens that change the outputs.” (D8ba)
- “versatility of DePass across various attribution granularities” (w5Wf)
- “Although conceptually related approaches exist, this work generalizes them to a broader setting” (w5Wf)

**Weaknesses:**

- Reviewers are dissatisfied with comparison to existing methods,
   - Reviewer D8ba would like to see more evaluation with sparse autoencoders (SAEs)
   - “omits evaluation against more mechanistic or causal interpretability methods such as activation patching, ablation-based attribution, or residual stream tracing” (w5Wf)
- Make notation more precise and add more examples (D8ba)

**Rebuttal:**

- The reviewers and authors engaged in significant back-and-forth discussion.
- Reviewer D8ba raised their score from 3->4 as a result, particularly noting the computational efficiency numbers provided in the rebuttal. “The authors' provided computational-efficiency suggested to me that this work at least occupies a place on the Pareto frontier of the expense-information density tradeoff of interpretability techniques”. They ended with, “I think the work is meaningful and worth accepting, particularly with the improved language and example coverage for clarity.”
- Despite significant discussion with the authors, Reviewer uE26 was unwilling to raise their score from 3. Going through the discussion, Reviewer uE26 continued to emphasize the advantages of SAEs as a mechanistic interpretability method, stating: “methods with some information loss but better downstream performance are more likely to be adopted in practice. Therefore, combining DePass with standard MI methods, including SAE, is important for this work.” They also mentioned a primary concern of theirs was computational efficiency. In response, the authors provide a detailed comment showing an example how their method can provide token-level attributions that align with SAE features.  Reviewer uE26 raised the concern this could be cherry-picked. The authors responded with more examples.
- Reviewer w5Wf maintained their original score of 4 after the rebuttal.

**Discussion:**

- Reviewer D8ba advocated for the paper, arguing that the method “seems cheap enough to run in parallel with other more expensive techniques, and that seems practically valuable”, and “Since the method does seem to pick up on relevant signal with reasonable efficiency, it seems to be on-margin useful.” With the caveat that the reviewer has uncertainty and that the paper as submitted was insufficient, so would have to be updated based on the rebuttal results.
- Reviewer uE26 reiterated their concern that the method fails to compare with existing techniques. They said, “the direction of the experiments is undoubtedly correct”, but that the rebuttal experiments are too preliminary to justify acceptance on the basis of what we’re able to review.

**Decision:**

Reviewers agree the method is novel, practical, and showing promising results. Specifically, that it adds value to the literature on mechanistic interpretability by providing a computationally less expensive method that still provides promising attribution results.

Because of Reviewer uE26’s dissenting opinion, the raw numerical scores would normally be below the threshold for acceptance. However, I disagree with Reviewer uE26’s insistence on the need to include results with SAEs, and rather think that the fact that this method is proposing an alternative to SAEs makes it more promising, given the number of works focusing on SAEs at the moment.

Therefore, my decision is to recommend acceptance. I strongly urge the authors to revise the paper based on the reviews, and in particular to include the new rebuttal results.